# Probiotics and Probiotic-like Agents against Chemotherapy-Induced Intestinal Mucositis: A Narrative Review

**DOI:** 10.3390/jpm13101487

**Published:** 2023-10-12

**Authors:** Laura López-Gómez, Alexandra Alcorta, Raquel Abalo

**Affiliations:** 1Department of Basic Health Sciences, Faculty of Health Sciences, University Rey Juan Carlos (URJC), 28922 Alcorcón, Spain; laura.lopez.gomez@urjc.es (L.L.-G.); alexandra.alcorta@urjc.es (A.A.); 2High Performance Research Group in Physiopathology and Pharmacology of the Digestive System (NeuGut-URJC), University Rey Juan Carlos (URJC), 28922 Alcorcón, Spain; 3Associated I+D+i Unit to the Institute of Medicinal Chemistry (IQM), Scientific Research Superior Council (CSIC), 28006 Madrid, Spain; 4Working Group of Basic Sciences on Pain and Analgesia of the Spanish Pain Society, 28046 Madrid, Spain; 5Working Group of Basic Sciences on Cannabinoids of the Spanish Pain Society, 28046 Madrid, Spain

**Keywords:** chemotherapy, mucositis, probiotic, synbiotic, paraprobiotic, postbiotic

## Abstract

Cancer chemotherapy has allowed many patients to survive, but not without risks derived from its adverse effects. Drugs, such as 5-fluorouracil, irinotecan, oxaliplatin, methotrexate, and others, as well as different drug combinations trigger intestinal mucositis that may cause or contribute to anorexia, pain, diarrhea, weight loss, systemic infections, and even death. Dysbiosis is a hallmark of chemotherapy-induced intestinal mucositis and diarrhea, and, therefore, strategies aimed at modulating intestinal microbiota may be useful to counteract and prevent those dreadful effects. This narrative review offers an overview of the studies performed to test the efficacy of probiotics and probiotic-like agents against chemotherapy-induced intestinal mucositis and its consequences. Microbiota modulation through the oral administration of different probiotics (mainly strains of *Lactobacillus* and *Bifidobacterium*), probiotic mixtures, synbiotics, postbiotics, and paraprobiotics has been tested in different animal models and in some clinical trials. Regulation of dysbiosis, modulation of epithelial barrier permeability, anti-inflammatory effects, modulation of host immune response, reduction of oxidative stress, or prevention of apoptosis are the main mechanisms involved in their beneficial effects. However, the findings are limited by the great heterogeneity of the preclinical studies and the relative lack of studies in immunocompromised animals, as well as the scarce availability of results from clinical trials. Despite this, the results accumulated so far are promising. Hopefully, with the aid of these agents, intestinal mucositis will be less impactful to the cancer patient in the near future.

## 1. Introduction

Currently, one of the main causes of death in the world is the development of different types of cancer. According to data provided by the Global Cancer Observatory [1], 19.3 million cases of cancer were diagnosed in 2020, and 9.96 million deaths occurred because of this disease. In addition, aspects associated with the current lifestyle such as diet, high consumption of ultra-processed foods, sedentary lifestyle, alcohol consumption, or smoking are significantly increasing the risk of suffering cancer [2]. For these reasons, in the forthcoming years, these statistics are expected to increase [1]. Fortunately, early detection and the advances in cancer treatments, make the survival rate of patients higher. However, these treatments have important side effects on patients, which can sometimes be highly limiting in their daily lives and, if not properly controlled, can lead to the cessation of treatment, even death.

Therapies for cancer vary depending on the disease type and stage. Some patients may receive more than one therapy, combining surgery, chemotherapy, and radiation. Chemotherapy acts mainly on rapidly dividing and proliferating cells, destroying these cancerous cells and avoiding their multiplication and division. However, during this process, healthy cells are also affected (particularly those with a high proliferating index), leading to undesirable and harmful side effects [3,4]. In particular, many drugs used for cancer treatments also induce apoptosis of healthy cells in the gastrointestinal tract, provoking mucosal damage (mucositis). Compounds as commonly used as 5-fluorouracil (5-FU) and its combinations with others (FOLFOX, FOLFIRI), irinotecan, methotrexate (MTX), capecitabine, cisplatin, oxaliplatin, or doxorubicin cause mucositis among other side effects [5]. 

Thus, chemotherapy affects the physical and functional intestinal barrier components, including the mucus layer, the epithelium, neuroendocrine feedback signaling, and/or immune system, as well as the gut vascular barrier. This may result in an immunological response and enhanced intestinal permeability to toxic substances and may facilitate the translocation of luminal bacteria to the underlying organs or systemic circulation [6]. Importantly, gastrointestinal mucositis is associated with malnutrition and harms the general state of health of cancer patients during treatment. It provokes diverse associated symptoms in them, such as the appearance of nausea and vomiting or diarrhea, and affects nutritional intake and intestinal function [7]. Indeed, mucositis constitutes a serious problem for the application of antitumor therapies, that sometimes must be even interrupted [8,9,10]. With this in mind, many researchers have been interested in the potential of natural products and nutraceuticals to alleviate the adverse effects of antitumor drugs, especially those related to the occurrence of mucositis in the gastrointestinal tract [4,11,12]. For this, the etiopathogenic mechanisms of chemotherapy-induced mucositis need to be understood.

As summarized in Figure 1, pathogenesis of mucositis has been defined as a five-step model. These overlapping steps are thought to be largely driven by the activation of nuclear factor-κB (NF-κB) [13,14,15]. Indeed, NF-κB plays an important role in regulating mucositis. Briefly, NF-κB is located in the cytoplasm of cells. When inactive, the classic form of NF-κB is tightly bound to the inhibitor of nuclear factor kappa B (IκB) class of proteins, which act as an ‘inhibitor’ of NF-κB function. Following cytotoxic therapy, the bound NF-κB/IκB is phosphorylated and ubiquitinated; then, NF-κB is allowed to enter the nucleus where it is able to up-regulate many genes associated with mucositis, including pro-inflammatory cytokines, growth factors, and pro- and antiapoptotic genes [16].

In advanced stages of mucositis, chemotherapy-induced pathological effects are observed macroscopically with the formation of ulcers, but there are also microscopic changes at cellular level such as alterations in mucin secretion. Mucins are important proteins, encoded by MUC genes, that contribute to the protection of the mucosa in the gastrointestinal tract. Mucins guarantee mucosa health against the luminal contents by regulating bacterial overgrowth and penetration, as well as providing attachment sites for commensal bacteria, and the disruption of this protection leads to an increased risk of infections in patients. Damage to epithelial cells is also related to the appearance of diarrhea, typically associated with mucositis. The capacity of water absorption of the gastrointestinal tract is reduced, and the enzymes located on the brush border of the villus such as maltase, sucrase, lactase, and aminopeptidase also decrease with cell loss. Unabsorbed substances accumulate in the lumen, creating an osmotic gradient. All this makes water move into the intestinal lumen contributing to diarrhea occurrence [6,16].

Of interest, patients receiving cytotoxic therapy exhibit marked changes in intestinal microbiota [13]. The microbiota produces short-chain fatty acids (SCFAs) such as butyrate, propionate, and acetate by metabolic processes that include the synthesis of vitamin K and several components of vitamin B, as well as the digestion and fermentation of carbohydrates that escaped proximal digestion and indigestible oligosaccharides. Trophic activities of the healthy microbiome involve controlling the integrity of the intestinal epithelium and ensuring immune system homeostasis. On the other hand, its protective role is related to the prevention of excessive proliferation of the resident pathogens [17]. 

Considering the key role of microbiota to maintain a healthy gastrointestinal mucosa, one of the most promising alternatives for the treatment of the adverse effects of chemotherapeutic drugs, particularly mucositis, may be the use of microbiota-related nutraceuticals, such as probiotics, synbiotics, postbiotics, or paraprobiotics [18] (see definitions in Table 1).

Probiotics are live microorganisms and, as such, they should have certain characteristics to exert maximum therapeutic effects and minimum risks for adverse effects. Thus, they should display resistance to the gastrointestinal tract environment (low pH and bile salts), because bacteria must remain viable to colonize the intestinal tract [17]. Despite their benefits on the host’s health, it is not clear if administering live microorganisms to immunosuppressed people may produce clinical problems due to bacterial translocation (bacteremia, endocarditis, liver abscess) [22]. These concerns would apply also to synbiotics and probiotic mixtures. In contrast, postbiotics or paraprobiotics, which are not live bacteria, would be less likely to cause harm to immunocompromised patients [23].

Probiotic properties widely differ between species, strains, or even between strain variants, which means these properties can be strain/variant specific [24]. Among the probiotics, those that have been most associated with the possibility of producing beneficial effects include the genera *Lactobacillus* and *Bifidobacterium*. These two genera include over 200 species among which many strains have been investigated as probiotics [17]. Various members of these genera are naturally associated with mucosal surfaces in the gastrointestinal tract, with mucosa protective actions [25]. 

The main mechanisms of action of probiotics to improve intestinal mucositis are summarized in Figure 2 [26].

In relation to these mechanisms, at the molecular level, it is worth mentioning that intestinal bacteria are tightly regulated by the Toll-like receptor (TLR) family. Gut microbiota is a source of TLR ligands [27]. Furthermore, TLRs participate in the regulation of NF-κB, the key regulator of alimentary mucositis [16]. Binding of bacterial products to TLRs on epithelial cells results in the activation of NF-κB signaling. Then, intestinal mucositis can activate the innate immune response through the activation of TLR-2 and TLR-4 leading to the development of an inflammatory response and to the release of proinflammatory cytokines (IL-6, IL-2, tumor necrosis factor (TNF)-α, and IL-1β). The release of IL-1β can activate the myeloid differentiation primary response 88 (MyD88) inflammatory cascade dependent of the Toll/IL-1R domain, inducing phosphorylation of the MAPKs ERK1/2, p38, and Jun N-terminal kinase (JNK). All this results in the modulation of various transcription factors, such as the activator protein-1 (AP-1), implied in important cellular events such as proliferation, differentiation, survival, and apoptosis [28]. As will be shown below, some probiotics are able to modulate these pathways, thus counteracting or preventing chemotherapy-induced intestinal mucositis and its consequences. 

Therefore, in the last years, the importance of probiotics and their derivatives is becoming more and more evident. Proof of this is the fact that in the Multinational Association of Supportive Care in Cancer and International Society for Oral Oncology (MASCC/ISOO) guidelines for the management of mucositis (a weighted summary of the best available scientific evidence, framed in a practical clinical context), the panel suggests that probiotics containing *Lactobacillus* spp. may be beneficial for the prevention of radiotherapy- or chemotherapy-induced diarrhea in patients with pelvic malignancy [14].

In this work, a narrative review has been carried out to gather all the available information on probiotics and related derivatives, which in recent years have attracted great interest in the treatment and prevention of chemotherapy-induced intestinal mucositis.

## 2. Methods

For the search of information, we have used the specialized health science database Pubmed. As inclusion criteria, we have considered both preclinical studies, using laboratory animals or cell cultures, and human trials, with no date limit and including both positive and negative results regarding the effectiveness of the probiotic or probiotic-like agents. As exclusion criteria, we have eliminated all articles that did not specifically deal with gastrointestinal mucositis (for example, those about oral mucositis, which are abundant) and we have focused only on chemotherapy treatments, excluding other methods for the treatment of cancer such as radiotherapy or immunotherapy.

Applying these criteria, we have found a total of 50 articles on the subject that have been analyzed to extract the most relevant information and present it as a narrative review. 

## 3. Probiotics and Mucositis

Probiotics are being extensively studied in the context of chemotherapy-induced mucositis, especially the genera *Lactobacillus* and *Bifidobacterium*, but also *Propionibacterium* or *Saccharomyces* have aroused interest due to their properties. Table 2 summarizes the main information collected on the research performed with these and other probiotics. A more detailed description of this research is provided in the text.

### 3.1. Lactobacillus Strains

#### 3.1.1. *Lacticaseibacillus casei*

Aragon et al. [29] analyzed the protective effect of milk fermented by the probiotic bacterium *Lacticaseibacillus casei* CRL 431 on a murine breast cancer model using female BALB/c mice. Mice were fed with milk fermented by *Lacticaseibacillus casei* 10 days before and 28 days after tumor injection. Probiotic administration delayed or blocked tumor development due to modulation of the immune response and reduced the area occupied by blood vessels in the tumors. This probiotic may have favorable effects against possible metastasis [29].

In a subsequent study by the same research group [30], mice bearing breast cancer were treated or not with capecitabine (500 mg/kg) and administered with the same probiotic fermented milk (PFM). PFM reduced capecitabine side effects and decreased intestinal mucositis and mortality. PFM administration also decreased diarrhea and maintained the villi length/crypt depth ratio similar to healthy animals in the small intestine. In addition, PFM by itself reduced metastasis and improved the host’s immune response. Interestingly, IL-6 was significantly reduced by PFM, and the reduction of this cytokine is related to cancer survival. Capecitabine treatment decreased levels of IL-10, a cytokine with immunosuppressive and anti-angiogenic functions, and PFM maintained this profile [30].

Innovative probiotic delivery strategies, based on probiotics incorporation into protective matrices, may increase its therapeutic effect by protecting bacteria against environmental stresses. Cordeiro et al. [31] tested the protection of whey protein isolate (WPI), when added to skim milk fermented by *Lacticaseibacillus casei* BL23, and the therapeutic effect of this fermented beverage (10^9^ CFU/mL) during 13 days in a murine model of mucositis induced by a single intraperitoneal injection of 5-FU (300 mg/kg). Milk supplementation with 30% (*w*/*v*) of WPI increases the survival rate of probiotic bacteria against the exposure to acid, bile salts, high temperature, and cold storage stresses. Moreover, treatment with the probiotic beverages prevented weight loss and intestinal damages in BALB/c mice receiving 5-FU. *Lacticaseibacillus casei* BL23 protective effect was further enhanced by the addition of WPI, which was able to decrease intestinal inflammation, preserve mucosal integrity, as well as prevent the degeneration of goblet cells and, consequently, improve protection and tissue repair. All these results suggest that presence of WPI maximizes the anti-inflammatory effects of this probiotic [31].

The recent study performed by Barbosa et al. [32] evaluated the effect of oral administration of *Lacticaseibacillus casei* for 18 days on the progression of 5-FU-induced intestinal mucositis (a single dose, 450 mg/kg) in female Swiss mice. *L. casei* reduced 5-FU-induced inflammation in the colon and small intestine and decreased the levels of TNF-α, IL-1β, IL-6 and malondialdehyde (a product of lipid peroxidation). In addition, a decreased expression of inducible nitric oxide (NO) synthase (iNOS) and TNF-α protein was found in the jejunum. The probiotic down-regulated NF-κB-P65 and TLR-4 gene expressions and up-regulated gene expression of the mucosal barrier proteins occludin and zonula occludens-1 (ZO-1) related with gut permeability. Furthermore, greater lactic acid bacteria population was found in the animals treated with *L. casei* compared with control groups, associated with normalization of intestinal microbiota, previously disrupted by 5-FU. Moreover, there was an increase in the mucin gene expression in the intestine of *L. casei* group. As already mentioned, the presence of mucins is important for the microbiota and protects the mucosa from bacterial overgrowth. On the other hand, *L. casei* was unable to protect against 5-FU-induced weight loss, probably due to associated anorexia and dehydration [32].

#### 3.1.2. *Lactobacillus delbrueckii*

*Lactobacillus delbrueckii subsp. lactis* CIDCA 133 fermented milk has been studied in 5-FU-induced experimental mucositis in male BALB/c mice. Animals received a single intraperitoneal injection of 5-FU (300 mg/kg). The probiotic was administered over a period of 13 days, 10 days prior and 3 days post 5-FU injection and subsequent mucositis induction. The probiotic strain CIDCA 133 significantly reduced 5-FU-induced shortening of small intestine length. This finding is important because a larger area of the intestine provides enough absorption surface for nutrient uptake and reduces the loss of water and electrolytes. For that reason, treated animals were able to recover the weight loss provoked by 5-FU. Treatment with CIDCA 133 fermented milk restored the loss of architecture of the ileum mucosa, greatly reduced the 5-FU-increased intestinal permeability, prevented loss of goblet cells (due to the protection of the stem cells inside the intestinal crypts), and reduced levels of neutrophil and eosinophil infiltration. Consequently, the severity of mucositis (measured as mucosal inflammation score) was reduced. Interestingly, the administration of probiotic fermented milk was able to decrease the levels of secretory IgA (sIgA). The level of sIgA is related to the intestinal barrier and is important in the maintenance of mucosal homeostasis and this reduction could be related to the improvement of intestinal mucosal barrier and the subsequent reduction of inflammation [33].

The same strain, under the same conditions, was investigated by Barroso et al. [34]. The probiotic provoked a decrease in the gene expression of TLR-2, TLR-4, Myd88, and NF-κB. In addition, the messenger ribonucleic acid (mRNA) expressions of proinflammatory cytokines (IL-6 and IL-1b) were downregulated while regulatory cytokine IL-10 was upregulated, further evidencing the anti-inflammatory effects that this strain can provide. Thus, CIDCA 133 could modulate inflammatory responses possibly by controlling NF-κB signaling pathway activation through upregulation of immunoregulatory molecules such as IL-10 to maintain intestinal homeostasis [34].

#### 3.1.3. *Lacticaseibacillus rhamnosus*

There are several studies, both in animal models of mucositis and in patients undergoing chemotherapy, which have used the species *Lacticaseibacillus rhamnosus*.

The sucrose breath test (SBT) can be employed to noninvasively assess the efficacy of probiotics, determining total intestinal sucrase activity. Sucrase is an enzyme found in the brush-border membrane of mucosal cells lining the lumen of the small intestine and catalyzes the breakdown of sucrose into its constituent monosaccharides, glucose and fructose. Following ingestion of 13C-sucrose, these monosaccharides are transported to the liver and metabolized to release ^13^CO_2_, which is exhaled from the lungs and quantified using an isotope ratio mass spectrometer. These in vivo determinations are indicative of mucosal damage, which is detected as a reduction in the parameter measured. Mauger et al. [35] used SBT to assess the efficacy of probiotics in 5-FU-induced intestinal mucositis in rats, but the probiotic was not effective to prevent it. 

Despite these early negative results, other authors did find positive effects on the administration of this probiotic. Thus, Yeung et al. [36] used BALB/c mice to investigate the effects of *Lacticaseibacillus rhamnosus* supplementation in ameliorating 5-FU-induced intestinal mucositis. Animals were injected with 5-FU (30 mg/kg/day for 5 days) intraperitoneally and gavaged with a probiotic suspension of *Lacticaseibacillus rhamnosus* (1 × 10^7^ CFU)/mL) daily. Diarrhea produced by 5-FU was attenuated following *Lacticaseibacillus rhamnosus* administration. TNF-α, IL-1β, and IL-6 mRNA expressions were up-regulated in intestinal tissues following 5-FU treatment and probiotic treatment suppressed this up-regulation, ameliorating inflammation. Histological alterations caused by 5-FU such as villus shortening, modification of crypt depth, and reduction in goblet cell numbers were restored by the probiotic [36].

Following the same protocol, Yeung et al. [37] also investigated the modulations of probiotics on gut microbiota. Stool specimens were collected for recombinant DNA (rDNA) extraction and pyrosequenced for bioinformatic analysis. At phylum and class levels of analysis, abundances of *Betaproteobacteria*, *Erysipelotrichi*, and *Gammaproteobacteria* were significantly increased by 5-FU. Probiotic supplementation did increase the abundances of *Enterobacteriales* and *Turicibacterales,* demonstrating that this strategy is capable of modulating microbiota composition [37]. The same group expanded this research using SCID/NOD mice to simulate the immunodeficiency of chemotherapy patients. SCID/NOD animals present a functional deficit of T and B cells, macrophage deficiency, and absence of circulating complement. The probiotic *Lacticaseibacillus rhamnosus* significantly improved the diarrhea scores in 5-FU-treated animals and TNF-α, IL-1β, and IL-6 serum levels were significantly inhibited. Importantly, no evidence of bacteremia in blood cultures was found [38].

Chang et al. [39] investigated the effect of *Lacticaseibacillus rhamnosus* on FOLFOX-induced mucosal injury. FOLFOX is a combined chemotherapy treatment including 5-FU, leucovorin, and oxaliplatin commonly used for the treatment of colorectal cancer. In this study, BALB/c mice subcutaneously injected with CT26 colorectal adenocarcinoma cells were orally administered *Lacticaseibacillus rhamnosus* daily before, during, and after 5-day injection of FOLFOX regimen, for 14 days. In this way, *Lacticaseibacillus rhamnosus* dose dependently reduced the severity of diarrhea and intestinal mucositis. Histological analysis of intestinal mucosa indicated that villus shortening, lengthening of the intestinal crypts, and reduction in the villus height-to-crypt depth ratio caused by FOLFOX was alleviated dose dependently by the probiotic. *Lacticaseibacillus rhamnosus* decreased FOLFOX-induced NF-κB activity in the intestine and improved mucositis, by suppression of inflammation. Probiotic administration reduced the upregulation of proinflammatory cytokines TNF-α and IL-6 in the jejunum. In this study, the probiotic significantly reduced FOLFOX-induced apoptosis of the intestinal crypt cells by reducing the increase in the BAX/Bcl-2 ratio and favoring anti-apoptosis vias. Taxonomic analysis at the phylum level indicated that FOLFOX changed the gut microbiota composition and significantly increased the abundance of *Firmicutes* and decreased the abundance of *Bacteroidetes*. These changes were restored by *Lacticaseibacillus rhamnosus* administration suggesting that the pathogenesis of mucositis is alleviated via the gut microbiota-TLRs-NF-κB signaling pathway [39].

Interestingly, this probiotic has also been tested in patients undergoing chemotherapy [40,41]. Patients diagnosed with colorectal cancer (*n* = 150) received monthly 5-FU and leucovorin bolus injections (the Mayo regimen) or a bimonthly 5-FU bolus plus continuous infusion (the simplified de Gramont regimen) for 24 weeks as postoperative adjuvant therapy. *Lacticaseibacillus rhamnosus* GG supplementation (1–2 × 10^10^ per day) reduced grade 3 or 4 diarrhea and abdominal discomfort, and patients needed less hospital care. Furthermore, no *Lacticaseibacillus*-related toxicity was detected [40]. In children with acute leukemia receiving *Lacticaseibacillus rhanmosus* supplementation (5 × 10^9^ CFU twice daily, by mouth), nausea, vomiting, and abdominal distension significantly decreased in the probiotic group [41].

#### 3.1.4. *Lactobacillus acidophilus*

Justino et al. [42] studied the effects of *Lactobacillus acidophilus* strain, a thermophilic nonpathogenic probiotic widely used for gastrointestinal disorders. The authors developed a model of intestinal mucositis induced by 5-FU administration (a single dose of 450 mg/kg) in male Swiss mice. *L. acidophilus* (16 × 10^9^ CFU/kg) was administered concomitantly with 5-FU on the first day, and for two additional days after administration. Mice with intestinal mucositis displayed significantly reduced villus height–crypt depth ratio and glutathione concentration but increased myeloperoxidase (MPO) activity and nitrite concentrations in jejunum and ileum. Furthermore, 5-FU significantly increased proinflammatory cytokines (TNF-α, IL-1β, and C-X-C Motif Chemokine Ligand 1 (CXCL-1)) concentrations, decreased IL-10 concentrations, delayed gastric emptying and gastrointestinal transit, and produced significant diarrhea. All these changes were significantly reversed by treatment with *L. acidophilus*, decreasing the inflammatory markers and functional aspects of intestinal mucositis induced by 5-FU [42].

#### 3.1.5. *Lactiplantibacillus plantarum*

Proinflammatory cytokines and reactive oxygen species (ROS) appear to be key factors during the pathogenesis of intestinal mucositis, and agents with anti-inflammatory/antioxidant activities may serve for the treatment or prevention of this adverse effect of chemo/radiotherapy. *Lactiplantibacillus plantarum* CRL2130 is a riboflavin-overproducing strain with anti-inflammatory properties. Riboflavin is widely known to act as an antioxidant and has a potential effect against oxidative stress. Mice were intraperitoneally injected with 5-FU (50 mg/kg) once a day for 6 days and, during this period, they received this probiotic twice daily (9 × 10^8^ CFU/mL). The administration of *L. plantarum* CRL2130 significantly attenuated the pathologic changes induced by 5-FU such as body weight loss, marked diarrhea, shortening of villus height, and the increase in the concentration of proinflammatory cytokines (IL-17, TNF-α, interferon (IFN)-γ, and IL-6 in serum and IL-17, TNF-α, IFN-γ, IL-6, IL-4, and IL-2 in intestinal contents). *Lactiplantibacillus plantarum* elevated the production of the anti-inflammatory cytokine IL-10, leading to an increase in the ratio of anti-/proinflammatory cytokines. These results indicate that the riboflavin-overproducing strain *Lactiplantibacillus plantarum* CRL2130 could be useful to prevent mucositis during cancer treatments [43].

#### 3.1.6. *Lactobacillus johnsonii*

MTX is an anti-metabolite that exerts its cytotoxic effect through the inhibition of folate metabolism by a down-regulatory effect on the enzyme dihydrofolate reductase, resulting in the inhibition of DNA synthesis. A study aimed to evaluate commercially available yoghurt products, containing *Lactobacillus johnsonii*, in adult male Sprague Dawley rats injected with 2.5 mg/kg/day for 3 days with MTX to induce intestinal mucositis. Although the cow’s yoghurt containing *L. johnsonii* (10^7^ organisms/L) was effective in reducing intestinal damage provoked by MTX, it was not as effective in restoring functionality as evidenced by brush-border enzyme activity, an indicator of damage in the small intestinal epithelium [44].

#### 3.1.7. *Lacticaseibacillus reuteri*

Yeung et al. [37] investigated the potential changes of 5-FU treatment and the modulations of probiotics on gut microbiota in male BALB/c mice. Animals were fed with 100 μL of a suspension containing 1 × 10^7^ CFU *Lacticaseibacillus reuteri* DSM 17938 daily and were administered with 5-FU (30 mg/kg/day) for 5 days. In the fecal microbial community, probiotics supplementation increased the abundances of *Enterobacteriales* and *Turicibacterales*. The authors concluded that the modulation of microbiota by administration of probiotics may be a useful strategy for the treatment of gastrointestinal side effects of chemotherapy [37].

#### 3.1.8. *Limosilactobacillus fermentum*

Not all *Lactobacillus*-based probiotics have been found to be effective in the treatment of mucositis. In the study performed by Mauger et al. [35] in Dark Agouti rats, this probiotic was administrated by gavage for 10 days (10^6^ CFU/mL) and intestinal mucositis was induced on day 7 by intraperitoneal injection of 5-FU (150 mg/kg). At the doses tested, *Limosilactobacillus fermentum* could not prevent the small intestine damage caused by 5-FU. Possible explanations for the lack of efficacy may be related to the dosage, timing, and duration of delivery, or lack of suitability of the strain used for the mucositis setting.

### 3.2. Bifidobacterium Strains

#### 3.2.1. *Bifidobacterium bifidum*


The study performed by Kato et al. [45] examined the effect of *Bifidobacterium bifidum* G9-1 (BBG9-1) on intestinal mucositis, especially in relation to its effects on apoptosis, inflammatory cytokine expression, and dysbiosis. In this investigation, mice received repeated administration of 5-FU (50 mg/kg) for 6 days. BBG9-1 (10^7^–10^9^ CFU) was administered orally once daily for 9 days, beginning 3 days before the onset of 5-FU treatment. Probiotic administration significantly attenuated body weight loss and reduced the severity of diarrhea induced by 5-FU. Furthermore, BBG9-1 abrogated the macroscopic and histological changes associated with 5-FU: reduction in small intestine length, shortening of the villi, and destruction of the crypts. Additionally, BBG9-1 significantly attenuated 5-FU-induced increase in MPO activity and TNF-α and IL-1β expression. Curiously, BBG9-1 failed to prevent the initial induction of apoptosis despite suppression of the secondary inflammatory responses. Interestingly, authors found that the clustering of the intestinal microbiota structure was altered by 5-FU, with a decrease in the abundances of *Firmicutes* and *Actinobacteria,* mostly comprised of Gram-positive bacteria, and increasing the abundances of *Bacteroidetes* and *Proteobacteria*, mostly Gram-negative bacteria. Treatment with BBG9-1 partially restored the normal composition of microbiota [45].

#### 3.2.2. *Bifidobacterium breve* Strain Yakult

In pediatric patients undergoing chemotherapy treatments against leukemia, the enteral administration of the probiotic *Bifidobacterium breve* strain Yakult was evaluated as a method to prevent infections. In immunocompromised patients, one of the main causes of infection is the alteration of endogenous intestinal flora and its colonization by pathogenic bacteria followed by translocation through the gut mucosa and systemic dissemination. For this study, a placebo-controlled trial was performed, and 42 patients received probiotic or placebo. The probiotic product contained 10^9^ freeze-dried, living *B. breve* strain Yakult. The frequency and duration of febrile episodes was lower in the probiotic group but there was no significant difference in the frequency and duration of diarrhea. Fecal bacteriological examination demonstrated the disruption of the intestinal microbiota after chemotherapy, with an increase in the population levels of *Enterobacteriaceae* more pronounced in the placebo group. Administration of *B. breve* strain Yakult resulted in a significantly higher count of *Clostridium leptum* subgroup after 2 weeks, indicating that the initiation of probiotic administration enhanced the intestinal occupation by anaerobes. Facultative anaerobes such as *Enterococcus* decreased only in the probiotic group. These results demonstrate its effect on maintenance of favorable intestinal microflora and suggest that administration of *B. breve* strain Yakult could be an effective approach for achieving clinical benefits in immunocompromised hosts by improving their intestinal environments [46].

#### 3.2.3. *Bifidobacterium infantis*

The benefits on mucositis of the probiotic *Bifidobacterium infantis* have been studied [47,48]. *B. infantis* is a commensal microbe isolated from the human gastrointestinal mucosa and has shown beneficial effects on gastrointestinal disease by modulating the immune function [48]. 

Male Sprague–Dawley rats were treated with a single intraperitoneal injection of 5-FU (150 mg/kg) and *B. infantis* (1 × 10^9^ CFU) was administered for 11 days, starting from 7 days before 5-FU injection. *B. infantis* improved body weight and reduced 5-FU-induced diarrhea. Animals treated with the probiotic restored villus height in jejunum, displayed increased expression of proliferating cell nuclear antigen (PCNA), reduced expression of NF-κB, and decreased cell damage. Pro-inflammatory factors and MPO concentration were decreased compared with the 5-FU group. These data suggest that the probiotic *B. infantis* is effective in reducing chemotherapy-induced intestinal mucositis and the occurrence of diarrhea in rats [47].

This probiotic has also proven to be useful in palliating mucositis caused by a combination of chemotherapeutic agents, 5-FU and oxaliplatin, due to its role in the regulation of T cell in a model of colorectal cancer in rats. In this model, animals were injected dimethyl hydrazine (DMH) subcutaneously for 10 weeks, and then injected with SW480 cells in the rectal mucosa. Chemotherapy was applied in these rats and a group of animals was treated with the probiotic [48]. The probiotic improved weight loss and restored intestinal villus height and crypt depth compared with the chemotherapy group. Interestingly, the authors demonstrated that this probiotic was able to modulate the activity of immune cells. During mucositis, effector T-helper cells (Th1, Th2 and Th17) released pro-inflammatory cytokines and initiated excessive inflammation, resulting in intestinal mucosal damage. This study showed that the levels of Th1 cells and derived cytokines (IL-2, IL-12, and IFN-γ) were up-regulated in the chemotherapy-induced intestinal mucositis rats and these effects could be reversed by *B. infantis* administration. *B. infantis* also down-regulated the level of T-bet (a Th transcription factor) that appears to regulate lineage commitment in CD4-Th lymphocytes, in part, by activating IFN-γ. In addition, the results showed a decrease in mRNA for the lineage-specific transcription factor RORγt in chemotherapy-induced intestinal mucositis in rats fed with *B. infantis*. The orphan nuclear receptor RORγt, is the pivotal transcription factor to Th17 differentiation. These results suggest that *B. infantis* suppressed Th17 responses by regulating its cytokines and differentiation-related factors. In this study, *B. infantis* promoted Foxp3^+^ Treg (regulatory T cells) responses in association with increased levels of IL-10 and transforming growth factor (TGF)-β in rats with intestinal mucositis. Foxp3^+^ Tregs are essential for normal immune homeostasis by suppressing T-helper effector cells. Overall, the data from this study seem to indicate that *B. infantis* can attenuate chemotherapy-induced intestinal mucositis via suppressing Th1 and Th17 responses as well as promoting Foxp3^+^ Treg responses in rats with colorectal cancer [48].

#### 3.2.4. *Bifidobacterium lactis*

SBT was employed to noninvasively assess the efficacy of probiotics in 5-FU-induced intestinal mucositis in an animal model using Dark Agouti rats. *Bifidobacterium lactis* BB12 was administered by oral gavage for 10 days and mucositis was induced on day 7 by intraperitoneal injection of 5-FU (150 mg/kg). Rats were sacrificed 72 h after 5-FU injection or vehicle (saline). In these experimental conditions, the probiotic offered no protection for mucositis at the dose tested [35].

### 3.3. Streptococcus thermophilus

Some studies have shown that the administration of *Streptococcus thermophilus* can reduce intestinal mucositis in animal models induced by chemotherapy agents such as 5-FU and MTX. This probiotic has been useful to prevent weight loss, attenuate diarrhea, and decrease intestinal damage. *S. thermophilus* is a Gram-positive, lactic acid producer, and ovoid-shaped bacterium appearing in pairs or in short chains. The capacity of production of SCFAs (acetate, propionate, and butyrate) may have positive effects for treatment of mucositis [54].

The study of Whitford et al. [49] analyzed the therapeutic potential of live *S. thermophilus* TH-4 (TH-4), dead TH-4, and TH-4 supernatant in rats treated with a single dose of 5-FU (150 mg/kg). Treatments were administered daily from two days before to three days after the administration of 5-FU, via orogastric gavage. Although in these conditions the probiotic was unable to ameliorate 5-FU-induced mucositis, live TH-4 may still have therapeutic potential in cancer patients, due to its capacity to decrease mitotic activity and reduce crypt fission and its potential to combat neoplasia [49].

Tooley’s research group [50] investigated the effects of orally ingested TH-4 on chemotherapy-induced small intestinal damage in female Dark Agouti rats using the noninvasive SBT test. Gastrointestinal damage was induced with MTX (1.5 mg/kg). Daily treatment with TH-4 at doses of 10^9^ (high), 10^8^ (low) CFU/mL, was performed 48 h pre- and 96 h post-MTX injection. Importantly, the administration of TH-4 at these two different doses produced differing results; when used at low doses, TH-4 offered no protection, whereas administration of TH-4 at the higher dose partially prevented the loss of body weight and the decrease in food intake and was capable of partially attenuating mucositis and normalizing SBT results. Again, these results highlight the importance of dosage in probiotic administration protocols to treat the side effects of chemotherapy [50].

To further investigate the therapeutic potential of this probiotic, the same research group carried out a study to analyze the effects of TH-4 on small intestinal damage and tumor progression in tumor-bearing rats [51]. Female tumor-bearing Dark Agouti rats (mammary adenocarcinoma) developed small intestinal damage induced via the administration of MTX (1.5 mg/kg) at 0 and 24 h. Animals were daily administrated with the high dose of TH-4 (10^9^ CFU/mL) −48 to +96 h post-MTX. When TH-4 was administered to MTX-treated rats with mammary adenocarcinoma, it yielded no protection. There could be several possible explanations for these findings. For example, the degree of intestinal damage was more severe in the tumor-bearing animals, compared with the damage observed in the non-tumor bearing model of mucositis [50]. Considering this probiotic was capable of increasing folate levels in reconstituted milk, the mechanism of protection by TH-4 in non-tumor bearing animals may have been the result of delivering a micro dose of folate to the site of damage. Nevertheless, the dose of 10^9^ CFU/mL was insufficient to diminish the severity of mucositis in the tumor presence [51].

The effects of this probiotic have also been studied with the antitumor drug doxorubicin (20 mg/kg) [52] in female Dark Agouti rats, gavaged with TH-4 (10^9^ CFU/mL) for 9 days. In this context, TH-4 partially prevented the loss of body weight associated with early induced mucositis development (24 h post-doxorubicin administration). This minimal amelioration may be due to folate production as a likely mechanism of TH-4 action against MTX-induced mucositis. Interestingly, in the saline-treated animals receiving TH-4, an increase in polymorphonucleocytes and lymphocytes in jejunum and ileum was observed. This could be indicative of an immunostimulatory response initiated through TH-4 administration alone. However, this effect was not observed in doxorubicin-treated rats receiving TH-4, most likely due to a masking effect provoked by the chemotherapeutic agent. Accordingly, it should be taken into consideration the potential for certain probiotics to impact on immune defenses. In general, the use of probiotics should be taken with caution, as adverse effects may occur and should be carefully analyzed. Further studies into TH-4 are required to confirm its applicability to chemotherapy regimens [52].

Levit et al. [53] evaluated two folate-producing strains, *S. thermophilus* CRL 808 and *S. thermophilus* CRL 415, against 5-FU-induced mucositis in mice. Mice were daily injected with 5-FU (50 mg/kg) to induce intestinal mucositis, and orally administered with folate-producing strains (100 ng/mL) during six days. *S. thermophilus* CRL 415 did not improve any of the parameters evaluated but *S. thermophilus* CRL 808 had the potential to prevent intestinal mucositis induced by 5-FU, decreasing diarrhea scores, improving histological alterations, and reducing jejunal inflammation. This effect was accompanied by decreased pro-inflammatory IL-6 and increased anti-inflammatory IL-10 in serum. The lack of a beneficial effect of *S. thermophilus* CRL 415 may be because this strain contains lower intracellular folate concentrations compared with *S. thermophilus* CRL 808. Therefore, the administration of selected folate-producing bacteria could be useful to attenuate the symptoms of intestinal mucositis in patients undergoing chemotherapy [53].

Shen et al. [54] studied if *S. thermophilus* ST4 separated from raw milk displays protective activity against intestinal mucositis induced by 5-FU (50 mg/kg/day). BALB/c mice received 100 μL of the probiotic suspension containing 5 × 10^8^ CFU, daily for 17 days. 5-FU intraperitoneal injection was administered after the 10th day, for three consecutive days. Administration of *S. thermophilus* alleviated diarrhea, decreased the infiltration of macrophage into distal mucosa and protected the structural integrity of small intestine and colon tissue. In addition, *S. thermophilus* ST4 significantly increased fecal acetic acid concentration, one of the common and functional SCFAs. SCFAs produced from bacterial fermentation may provide energy for epithelial cells essential for the development and mediation of the intestinal barrier function. Proinflammatory cytokines TNF-α, IL-1β, and IL-6 were significantly increased following 5-FU treatment, and this effect was attenuated by *S. thermophilus*. In addition, TNF-α is a key factor in the caveolin-1-mediated internalization of occludin and, therefore, its reduction improved gut permeability [54].

### 3.4. Propionibacterium freudenreichii 

*Propionibacterium freudenreichii* is an important species of dairy propionibacteria that plays an important role in food transformation, particularly in cheese ripening. It is a Gram-positive, non-motile, non-spore forming, and anaerobic to aerotolerant beneficial bacterium that is generally recognized as safe (GRAS). Dairy propionibacteria have a great probiotic potential, as they produce important metabolites such as the SCFAs acetate and propionate, vitamins B9 and B12, as well as 1,4-dihydroxy-2-naphthoic acid (DHNA) and 2-amino-3-carboxy-1,4-naphthoquinone (ACNQ) [31].

Only a few studies have tested the immumodulatory properties of this probiotic. *P. freudenreichii* CIRM-BIA 129 strain possesses extractable surface proteins, including surface-layer protein (Slp) B, involved in anti-inflammatory effect and in adhesion to epithelial cells. The work by Do Carmo et al. [55] studied the importance of SlpB in *P. freudenreichii* ability to reduce mucositis inflammation both in vitro and in vivo, comparing *P. freudenreichii* wild type with a mutated strain of *P. freudenreichii* ΔslpB, lacking the SlpB protein. First, in an in vitro assay, *P. freudenreichii* WT reduced the expression of proinflammatory cytokines IL-8 and TNF-α in lipopolysaccharide (LPS)-stimulated HT-29 cells, but *P. freudenreichii* ΔslpB, lacking the SlpB protein, failed to do so. Hereafter, both strains were investigated using an in vivo model in BALB/c mice, after induction of mucositis by a single intraperitoneal injection of 5-FU (300 mg/ kg). The probiotics were administered daily for 10 days before 5-FU injection. Wild-type strain prevented weight loss, reduced inflammation and consequently histopathological scores. The claudin-1 expression, decreased in mucositis, was restored by consumption of the probiotic, but not by the mutant, in accordance with its inability to restore gut permeability. Moreover, the wild type of the probiotic produced a decrease in sIgA levels that could be correlated with the improved integrity of the epithelial barrier and consequent protection against pathogens. Altogether, the findings of this work demonstrated, by in vitro and in vivo approaches, that the mutation of the extractable surface protein slpB gene directly affects the probiotic effects of *P. freudenreichii* [55].

*P. freudenreichii* CIRM-BIA138 (*P. freudenreichii* 138) has been tested in combination with a protective whey matrix (WPI) to reduce the effects of bacteria exposure to stressful environments and increase their probiotic properties. Although a probiotic beverage fermented by *P. freudenreichii* 138 was able to decrease 5-FU-induced intestinal inflammation in BALB/c mice, with preservation of the mucosal integrity and reduced weight loss, the presence of WPI did not improve these effects. In contrast, as mentioned above, *L. casei* BL23 protective effect was further enhanced by the addition of WPI. These results suggest that whey protein enhancement of probiotic is strain dependent [31].

### 3.5. Clostridium butyricum 

*Clostridium butyricum* is a butyrate-producing human gut symbiont. *C. butyricum* consume undigested dietary fibers and generate SCFAs, specifically butyrate and acetate. In addition, *C. butyricum* may modulate the composition of the gut microbiome, possibly increasing certain beneficial bacterial taxa such as *Lacticaseibacillus* and *Bifidobacterium* [62].

The role of *C. butirycum* in patients undergoing chemotherapy was investigated in the study performed by Tian et al. [56]. A total of 41 participants with lung cancer subjected to platinum-based chemotherapy were administered with *C. butyricum* or placebo for 3 weeks. The incidence of chemotherapy-induced diarrhea was lower in the group treated with the probiotic and the systemic inflammatory response system was reduced. *Firmicutes* in the placebo group decreased significantly; however, in the *C. butyricum* group, the variation was minor. The genera producing SCFA tended to increase and the pathogenic genera tended to decrease in the *C. butyricum* group and a notable increase in beneficial flora was detected, including *Clostridium* and *Lacticaseibacillus*. The present study highlighted that *C. butyricum* reduced chemotherapy-induced diarrhea in patients with lung cancer, helping to maintain the condition of the intestinal flora [56].

### 3.6. Saccharomyces Strains

*Saccharomyces cerevisiae* is a brewer’s, baker’s yeast with probiotic properties. Several taxonomic studies on *S. boulardii* have indicated that it should be considered as a strain of *Saccharomyces cerevisiae* and, as such, it should be referred to as *Saccharomyces cerevisiae var. boulardii* [63]. However, other studies suggest that they are different enough to be considered separate species. Despite the striking relatedness in molecular phylogeny and typing, *S. boulardii* does possess identifiable distinct traits and is physiologically and metabolically distinct from *S. cerevisiae*. *S. boulardii* is incapable of producing ascospores, switching to haploid form, or using galactose as carbon source. It is more resistant to temperature and acidic stresses, but less resistant to bile salts [25]. Whatever the case may be, until now, *Saccharomyces boulardii* has been the only yeast commercialized worldwide as a probiotic for humans [64].

#### 3.6.1. *Saccharomyces cerevisiae*

To evaluate the potential probiotic effects of *Saccharomyces cerevisiae* UFMG A-905 (Sc-905), Bastos et al. [57] investigated its effects in a mucositis model induced by irinotecan (75 mg/kg) in Swiss male mice. For this, they tested different administration possibilities: pre- or post-treatment with viable or inactivated yeast for 5 days. Their study included the effects of Sc-905 cells killed by heat, due to the possibility of some risk for immunocompromised patients when using live probiotics. In this study, only post-treatment with viable Sc-905 (0.1 mL of 10^9^ CFU/mL) was able to protect mice against the damage caused by chemotherapy. Histological analysis of the jejunum showed the improvement of the architecture of intestinal mucosa, reduction of the mucosal inflammation, and prevention of loss of goblet cells in animals treated with irinotecan; furthermore, this treatment was able to reduce the alteration of intestinal permeability. Sc-905 yeast reduced the damage caused by the formation of ROS during chemotherapy, such as lipid peroxidation, and was shown to provide greater availability of glutathione (GSH), which quenches ROS [57]. 

There is an increased interest in yeasts enriched with selenium as a dietary supplement, which may protect against oxidative stress due to their ability to incorporate inorganic selenium and convert it into selenomethionine [58]. Oral administration of Sc-905 enriched with selenium was evaluated as an alternative to alleviate the side effects of 5-FU-induced mucositis in Swiss mice. The researchers administrated a single intraperitoneal injection of 5-FU (300 mg/kg) and a daily dose of 10^8^ CFU of Sc-905 enriched (or not) with selenium for 10 days. Both probiotics demonstrated positive effects, but selenium-enriched yeast proved to be more effective to reduce MPO activity, levels of the neutrophil chemoattractant cytokine CXCL1/KC, histopathological tissue damage, oxidative stress (lipid peroxidation and nitrite production), and the increase in NO levels associated with the induction of mucositis. However, only selenium-enriched yeast reduced eosinophil peroxidase activity produced by 5-FU in the small intestine Altogether, this study clearly showed that the oral administration of Sc-905 protected mice against mucositis induced by 5-FU, and that this effect was potentiated when the yeast was enriched with selenium [58].

#### 3.6.2. *Saccharomyces boulardii*

The thermophilic non-pathogenic yeast *Saccharomyces boulardii* has been investigated on irinotecan-induced mucositis and diarrhea in male Sprague–Dawley rats. Irinotecan (60 mg/kg) was administered intravenously for four consecutive days and *Saccharomyces boulardii* (800 mg/kg) was administered for 3 days before administration of irinotecan and 7 days throughout the experiment. In the jejunum, leukocyte infiltration and inflammation were significantly lower in rats which received the probiotic. This yeast prevents intraluminal solute accumulation and the subsequent osmotic diarrhea by increasing the enzymes in the brush border in chemotherapy-exposed animals. Moreover, *S. boulardii* contains significant amounts of polyamines (spermidine and spermine). In the small intestine, these polyamides are absorbed by semi-active transport system; via polyamines, *S. boulardii* has effects on cell growth, differentiation, maturation, and apoptosis, controlling mitogen-activated protein (MAP) kinase (MAPK) and NF-κB. The synthesis of the proinflammatory cytokines TNF-γ and IL-8 are controlled by MAPK and NF-κB. In this way, *S. boulardii* modifies the signaling pathways implicated in the synthesis of proinflammatory cytokines and contributes to reducing irinotecan-induced mucosal damage [59].

Justino and collaborators have conducted several studies on the efficacy of this probiotic in relieving mucositis caused by a single intraperitoneal administration of 5-FU (450 mg/kg) in a murine model [28,60]. Animals were treated with *S. boulardii* (16 × 10^9^ CFU/kg) for 3 days. *S. boulardii* reversed 5-FU-induced changes in gastrointestinal function, enhancing intestinal transit and gastric emptying and attenuating diarrhea and weight loss. In parallel, this probiotic induced the recovery of intestinal permeability measured by lactulose: mannitol ratio. In jejunum and ileum, *S. boulardii* significantly reversed the histopathological changes in intestinal mucositis and reduced the inflammatory parameters such as neutrophil infiltration, MPO activity, nitrite concentration, proinflammatory cytokines TNF-α and IL-1β, and also reduced GSH concentration. In addition, treatment with *S. boulardii* significantly decreased apoptosis in intestinal crypt cells in the jejunum and ileum. Furthermore, it modified the Toll-like/MyD88/NF-κB/MAPK pathway in an infection model of Caco2 cells exposed to LPS and treated with 5-FU [60]. Similar results were observed in 5-FU-mediated experimental intestinal mucositis in mice. The pharmacological modulation of this pathway by *S. boulardii* might have a relevant therapeutic impact [28].

Despite promising results on the use of this yeast to alleviate the mucositis produced by chemotherapy, some results have been contradictory. In the study performed by Maioli et al. [61], pretreatment for 10 days with *S. boulardii* (10^9^ CFU/mL) was not able to prevent the effects of experimental mucositis induced by 5-FU (300 mg/kg) such as the increase in intestinal permeability or alterations in the tissue architecture [61].

## 4. Probiotic Mixtures and Mucositis

To enhance the beneficial effects of probiotics, some authors have used mixtures of different bacteria to treat mucositis and its symptoms. Different strains of *Lacticaseibacillus* and *Bifidobacterium*, as well as *Streptococcus thermophilus*, have been combined in this context (Table 3).

A mixture with probiotics *L. bulgaricus* and *S. thermophilus* has been tested to see if it improves small intestinal permeability in a model of MTX-induced mucositis. Southcott et al. [44] used commercially available yoghurts containing this mixture of probiotics. Male Sprague–Dawley rats were gavaged twice daily for 7 days pre-MTX and 5 days post-MTX. The probiotic mixture improved small intestinal barrier function determined by a decreased lactulose/mannitol ratio. In addition, it was able to improve brush-border sucrase and lactase activity and tissue architecture altered by MTX [44].

The study performed by Tang et al. [65] analyzed the effects of a probiotic mixture, DM#1, on intestinal mucositis and dysbiosis in male Sprague–Dawley rats treated with intraperitoneal injections of 5-FU (30 mg/kg) for 5 days. DM#1 is composed by four probiotic strains: *Bifidobacterium breve* DM8310, *L. acidophilus* DM8302, *L. casei* DM8121, and *S. thermophilus* DM8309. This mixture was intragastrically administered for 8 days in different concentrations: high (10^9^ CFU/kg) and low (10^8^ CFU/kg). DM#1 mixture ameliorated 5-FU-induced dysbiosis, restoring *Lactobacillus* spp. and *Clostridium* clusters III and XIVa. The Clostridium XIVa group may attenuate intestinal inflammation by exerting an effect on polyamine secretion, which in turn regulates the expression of TLR-2. Administration of the high dose of the probiotic mixture to 5-FU-treated rats attenuated the increase in TLR-2 mRNA expression and, consequently, reduced the levels of proinflammatory cytokines TNF-α and IL-4 and IL-6 and intestinal permeability [65].

The same probiotic mixture was evaluated on cisplatin-induced mucositis and pica (an indirect maker of emesis in rats) [66] by daily intraperitoneal injection of cisplatin (6 mg/kg) for 3 days. A high dose (1 × 10^9^ CFU/kg) of the probiotic mixture was administered daily for 1 week before and 6 days after cisplatin injection. Administration of cisplatin produced changes in microbiota, increasing the relative abundances of *Enterobacteriaceae*, *Blautia*, *Clostridiaceae*, and members of *Clostridium* clusters IV and XIVa, which produce butyrate and thereby promote the secretion of the neurotransmitter 5-hydroxytryptamine (5-HT, serotonin) by enterochromaffin cells. In addition, 5-HT interacts with 5-HT_3_ receptors on vagal afferent terminals in the wall of the bowel and transmits the stimulus to the brain, resulting in emesis. 5-HT also regulates interstitial cells of Cajal (ICCs) in the gastrointestinal tract and aquaporin 3 expression in the colon, which is closely related to constipation and diarrhea. The administration of the probiotic mixture ameliorated cisplatin-induced mucositis and pica in rats partly by normalizing the dysbiosis-driven 5-HT overproduction [66].

Quaresma et al. [67] evaluated the effects of two different probiotic mixtures on intestinal mucositis. Swiss male mice were administered with 5-FU (450 mg/kg) and treated with the mixtures. PM-1 was a mixture of *L. acidophilus* and *B. lactis*, whereas PM-2 contains *L. acidophilus*, *Lacticaseibacillus paracasei*, *Lacticaseibacillus rhamnosus*, and *B. lactis*. Both probiotic mixtures had positive effects on various aspects related to mucositis. They reversed the structural changes caused by 5-FU, caused an increase in the villus height/crypt depth ratio of the intestinal mucosa, previously reduced by 5-FU, and a decrease in histopathological scores. PM-1 and PM-2 also reduced neutrophil infiltration and the production of proinflammatory cytokines TNF-α and IL-6. Both mixtures reduced GSH consumption in the duodenum and jejunum, but not in the ileum. In addition, the increased lipid peroxidation produced by 5-FU was reversed in the three intestinal segments, in line with the reduction in GSH consumption. In the other hand, chemotherapy caused loss of body weight and this effect was only prevented by treatment with PM-2, maybe due to effects of the two additional species that it contains (*Lacticaseibacillus rhamnosus* and *paracasei).* Finally, 5-FU administration induced a considerable delay in gastric emptying during the inflammatory phase that was only partially reversed by treatment with PM-2. Thus, the choice of the probiotics used in the different mixtures seems to be crucial, because the results obtained are different depending on the strains used [67].

Other two studies, both performed by the same research group, evaluated the effects of the combination of *L. acidophilus* and *B. bifidum* (LaBi) in mucositis induced by 5-FU administration in BALB/c mice, with a normal immune system [36] and NOD/SCID mice, to simulate the immunodeficiency status of cancer [38]. Marked diarrhea developed in the 5-FU groups but it was attenuated after oral LaBi administrations in both types of animals, and the probiotic mixture also suppressed the upregulation of cytokines (TNF-α, IL-6, and IFN-γ). Adding LaBi restored villus height in BALB/c mice, but this change was not significant in NOD/SCID mice. It is possible that increased T lymphocyte activity in BALB/c enables stronger protection and healing of mucosal villi. In BALB/c mice, no bacteremia was noted when bacterial translocation was evaluated in samples of blood, liver, and spleen [36]. In contrast, in the spleen and liver samples as well as mesenteric lymph node samples from NOD/SCID mice, LaBi strains were detected via polymerase chain reaction (PCR), but no bacteremia was found in blood samples [38].

Bowen et al. [68] investigated the commercial probiotic mixture, VSL#3, for amelioration of chemotherapy-induced diarrhea in female Dark Agouti rats. In this case, irinotecan was used to induce mucositis and diarrhea. VSL#3 (VSL Pharmaceuticals, Italy) contains concentrated freeze-dried living bacteria (300 billion viable bacteria per gram), including four strains of lactobacilli (*Lactiplantibacillus plantarum, Lacticaseibacillus casei, L. acidophilus*, and *L. delbrueckii* subspecies *bulgaricus*), three strains of bifidobacteria (*B. infantis*, *B. longum*, and *B. breve*), and one strain of streptococcus (*Streptococcus salivarius* subspecies *thermophilius*). VSL#3 reduced weight loss following irinotecan when administered before and after chemotherapy and prevented moderate and severe diarrhea. This improvement seems to be due to a combination of beneficial effects: in jejunum and colon, this mixture increases epithelial proliferation and healing of the mucosal layer, reduces intestinal apoptosis helping to prevent gastrointestinal damage, and prevents the increase in goblet cell number and mucin secretion in jejunal crypts, thus helping to maintain water and electrolyte balance within the intestine and consequently preventing the onset of diarrhea [68]. 

More recently, a mixture of selected lactic acid bacteria (LAB) composed by a riboflavin-overproducing strain (*Lactiplantibacillus plantarum* CRL 2130), a folate-producing strain (*S. thermophilus* CRL 808), and an immunomodulatory strain (*S. thermophilus* CRL 807) was used by Levit et al. [69] in an animal model of breast cancer in female BALB/c mice treated with 5-FU. LAB mixture decreased tumor growth and showed modulation of systemic cytokines modified by both tumor growth and 5-FU treatment. Mice treated with the probiotic mixture showed less damage at intestinal structure level and a reduction of the degree of diarrhea. The anti-inflammatory potential of the bacterial mixture decreased serum concentrations of cytokines IL-6 and TNF-α. Increased levels of the regulatory cytokine IL-10 were detected, associated with the immunomodulatory properties of *S. thermophilus* CRL 807. The LAB mixture attenuated the intestinal mucositis associated with 5-FU treatment [69].

Another multispecies probiotic mixture composed by eight bacterial strains (*Lacticaseibacillus casei* W56; *Lactobacillus acidophilus* W37; *Levilactobacillus brevis* W63; *Lactococcus lactis* W58; *Bifidobacterium lactis* W52; *Lactococcus lactis* W19; *Ligilactobacillus salivarius* W24; and *Bifidobacterium bifidum* W23) combined with 1 g of matrix (vegetable proteins, potassium chloride, amylases, maize starch, maltodextrins, magnesium sulphate, and manganese sulphate) or placebo was dissolved in tap water and administered by gavage to male Wistar rats for two weeks. Animals were implanted or not with a tumor and exposed to chemotherapy with FOLFOX. The use of the probiotic mixture alleviated the severity, duration of diarrhea, and the intestinal damage induced by chemotherapy. In addition, probiotic supplementation increased intestinal cell proliferation and reduced apoptosis of cells in the ileum, but not in colon samples. All these results indicate the ability of this combination of probiotics to alleviate some mucosal damage and reduce related side effects, such as diarrhea [70].

Some probiotic mixtures have been tested in patients. For example, *Bifidobacterium tetragenous* viable bacteria tablets, composed by *B. infantis*, *L. acidophilus*, *Enterococcus faecalis*, and *Bacillus cereus*, were investigated to treat functional constipation during chemotherapy in a cohort of 100 cancer patients. The treatment lasted 4 weeks and in 48 patients constipation was controlled by the probiotic mixture [71]. In another group of 46 patients with colorectal cancer receiving irinotecan-based therapy, the probiotic formula Colon Dophilus™ was administered at a dose of 10 × 10^9^ CFU of bacteria orally for 12 weeks. Each capsule contained 10 lyophilized probiotic strains including *B. breve*, *B. bifidum*, *B. longum*, *Lacticaseibacillus rhamnosus*, *L. acidophilus*, *Lacticaseibacillus casei*, *Lactiplantibacillus plantarum*, *S. thermopilus*, *Levilactobacillus brevis*, and *B. infantis*. Administration of this formula led to a reduction in the incidence of severe diarrhea of grade 3 or 4 as well as of the incidence of enterocolitis [72].

## 5. Synbiotics and Mucositis

The combination of a probiotic and a prebiotic is known as a synbiotic and seeks to obtain a greater biological effect than probiotics administered individually. Administration of synbiotics appears to be an alternative therapy to treat or prevent mucositis (Table 4).

Smith et al. [73] studied the possibility of using *Limosilactobacillus fermentum* BR11 and the prebiotic fructo-oligosaccharide (FOS) in synbiotic combination to alleviate intestinal mucositis provoked in female Dark Agouti rats by a single injection of 5-FU (150 mg/kg). In this case, the probiotic doses used were 10^6^ or 10^9^ CFU/mL. In the small intestine, the combination of BR11 and FOS did not provide additional protection than the probiotic alone. Moreover, the high dose of the synbiotic increased the severity of 5-FU-induced mucositis. Authors concluded that, although *Limosilactobacillus fermentum* BR11 has the potential to reduce inflammation of the upper small intestine, its combination with FOS does not appear to confer any therapeutic benefit for the alleviation of mucositis [73].

The effect of mulberry leaf extract fermented with the probiotic strain *L. acidophilus* A4 administered for 10 days was evaluated on intestinal mucositis induced by 5-FU (150 mg/kg) in male Wistar rats. Mulberry leaves contain many phenolic compounds such as gallic acid, catechin, gallocatechin gallate, caffeic acid, epicatechin, rutin, resveratrol, and quercetin. As expected, 5-FU-induced significant weight loss, shortened villi height, and increased histological severity, IL-1β expression, and MPO activity. These pathological alterations were ameliorated by treatment with the probiotic alone, mulberry leaf alone, and mulberry leaf fermented with the probiotic. It should be noted that fermented mulberry leaf significantly reduced neutrophil infiltration in the small and large intestine, consequently resulting in less inflammation, providing greater protective effect. The treatment had positive effects on protective mucin production by stimulating expression of mucin genes MUC2 and MUC5AC. Authors conclude that fermented mulberry leaf had greater protective effect than the use of the probiotic *L. acidophilus* A4 alone on 5-FU-induced mucositis in rats [74].

Takahashi et al. [75] developed a synbiotic, Gut Working Tablet (GWT), which was tested to alleviate the symptoms of jejunoileal MTX-induced mucositis in rats. GWT is composed of the fermentation products of several cereal germs with the *Aspergillus oryzae* strain NK (*A. oryzae* NK-Koji) (a lactic acid bacterium), *Enterococcus faecium* and its fermentation products, and *S. cerevisiae*. In this study Sprague–Dawley rats were injected with MTX (7.5 mg/kg) for 4 consecutive days and fed for 3 weeks with a commercial chow powder, containing 5% GWT. Administration of GWT restored intestinal integrity by reversing villi shortening, crypt loss, and goblet cell depletion in the mucosa. GWT reduced activities of MPO and lipid peroxidase, increased superoxide dismutase activity, and suppressed mRNA expression of TNF-α and IL-12 in macrophages, demonstrating the protective role of GWT in chemically induced alterations produced by MTX in jejunoileal mucosa [75].

Oral administration of a synbiotic (Simbioflora^®^) preparation containing *Lacticaseibacillus paracasei*, *Lacticaseibacillus rhamnosus, L. acidophilus,* and *B. lactis* plus FOS was evaluated by Trindade et al. [76] to control experimental mucositis induced by a single intraperitoneal injection of 300 mg/kg 5-FU in male BALB/c mice. Animals received the synbiotic daily by oral gavage for 13 days. The synbiotic significantly maintained the villus and crypt height and reduced the inflammatory infiltration in the small intestine. Staining with periodic acid–Schiff (PAS) indicated that animals treated with the synbiotic had a thicker mucus layer. Mucositis increased MPO activity in the small intestine, but this change was not reversed by the synbiotic. In contrast, eosinophil peroxidase activity was reduced by treatment with the synbiotic, reflecting a reduction of inflammatory cell activity. It is worth mentioning that there was an increase in the concentration of SCFAs acetate, butyrate, and propionate in the feces of mice with mucositis treated with the synbiotic. These compounds have a direct influence on regulatory cells and inflammation. The metabolism of SCFA induces the stimulation of G-protein-coupled receptor 43 (GPR43), which, by downstream signaling, induces peroxisome proliferator-activated receptor (PPAR)-γ activation, which blocks NF-κB translocation to the nucleus. The results indicate that administration of this synbiotic can decrease mucosal damage caused by mucositis [76].

Savassi et al. [77] developed a lyophilized synbiotic product, containing skimmed milk, supplemented with WPI and with FOS, and fermented by *Lacticaseibacillus casei* BL23, *Lactiplantibacillus plantarum* B7, and *Lacticaseibacillus rhamnosus* B1. In a mice 5-FU mucositis model (female BALB/c mice, a single intraperitoneal injection of 5-FU, 300 mg/kg), this synbiotic reduced animal weight loss. The synbiotic formulation presented anti-inflammatory activity, controlling the levels of pro-inflammatory cytokines IL-1β, IL-6, IL-17, and TNF-α in the ileum and reducing mucosal damage caused by chemotherapy. The fermentation processes produced by the three strains and the production of metabolites (SCFA, bacteriocins, and neurotransmitters such as GABA) may be the reason for this modulatory effect. In addition, this formulation was able to reduce intestinal permeability as the synbiotic product promoted an increase in the expression of genes involved in the intestinal epithelial barrier (ZO-1, occludin, and claudin-1) [77].

## 6. Postbiotics and Mucositis

Probiotic-derived supernatants have also been investigated in models of mucositis due to their capacity to carry out similar functions to the live bacteria from which they are derived, with a reduced risk of infection in immunocompromised patients. The supernatants may contain compounds (SCFAs or bacteriocins) beneficial for treatment of mucositis [78]. For this purpose, the supernatants derived from *S. thermophilus* TH-4, *Faecalibacterium prausnitzii* (Fp), *Limosilactobacillus fermentum*, and *Escherichia coli* Nissle 1917 have been tested for the treatment of 5-FU-induced mucositis (Table 5).

Whitford et al. [49] evaluated the effects of live *S. thermophilus* TH-4 supernatant in rats treated with 5-FU (150 mg/kg). The authors found that it increased crypt depth and improved crypt fission, suggesting a therapeutic utility of TH-4 supernatants in the prevention of disorders such as colorectal carcinoma [49].

The therapeutic ability of supernatants from *Escherichia coli* Nissle 1917 (EcN) and *Limosilactobacillus fermentum* BR11 (BR11) to decrease 5-FU-induced damage in female Dark Agouti rats was investigated by Prisciandaro et al. [78]. Mucositis was induced by an intraperitoneal injection of 5-FU (150 mg/kg) on day 5, and animals were gavaged with 1 mL of either the supernatant or vehicle daily for 8 days. However, the supernatant was not able to improve all measured indicators of 5-FU-induced damage and only partially protected the intestine as indicated by reduced histological severity scores and improved intestinal morphology. It is worth noting that MPO was increased in the jejunum of rats treated with BR11 supernatant + 5-FU. However, in EcN supernatant + 5-FU-treated rats, the levels of MPO were reduced probably due to a reduction of TNF-α levels [78].

Wang et al. [79] reported the effects of *Faecalibacterium prausnitzii* (Fp) and EcN supernatants in 5-FU-treated intestinal cells. Cells cultured in vitro were analyzed for monolayer permeability by transepithelial electrical resistance (TER). Among the three cell lines tested (IEC-6, Caco-2, and T-84), Fp supernatant enhanced barrier integrity and prevented the barrier disruption induced by 5-FU in Caco-2 and T-84 cells while having no such effect on IEC-6 cells. In a rat model (female Dark Agouti rats) of mucositis provoked by a single injection of 5-FU (150 mg/kg), the same supernatants were administered to the animals once daily via orogastric gavage for 8 days and partly prevented body weight loss, normalized water intake, and restored crypt depth in the jejunum. However, they failed to alleviate the increased MPO levels induced by 5-FU. Factors derived from the probiotics present in the supernatants may be responsible for reducing the severity of intestinal mucositis [79].

In general, these studies suggest less effectiveness of probiotic derivates than living probiotic bacteria, and further studies are needed to analyze how to increase their efficacy for therapeutic use against the side effects of chemotherapy, particularly in immunocompromised patients.

## 7. Paraprobiotics and Mucositis

Studies on paraprobiotics are still very scarce, but the results obtained thus far seem promising (Table 6).

In the only study found to illustrate this topic, performed by Nobre et al. [80], C57BL/6 male mice received intraperitoneal injections of irinotecan (75 mg/kg), once a day for 4 days for mucositis induction. Paraprobiotic EC-12^®^ (*Enterococcus faecalis*, 5 × 10^9^ CFU/g) or paraprobiotic-based preparation Med Lan–S^®^ (pasteurized *Enterococcus* and *Bifidobacteria*, dextrin, soluble guar gum, maltose syrup, lemon juice powder, starch/citric acid, and stevia sugar) were administered for one week before mucositis induction. Interestingly, bacteremia was evaluated to determine if the administration of these products was safe. Both paraprobiotics improved mucositis-related aspects. They attenuated the neutrophil infiltration in the intestine and increased the villus/crypt ratio. Additionally, they reduced the mRNA expression of claudin-2 and occludin but only administration of EC-12 prevented the increased expression of IL-18, IL-18BP, and TLR-4. Remarkably, EC-12 and Med Lan inhibited the irinotecan-induced bacterial translocation to the blood associated with mucositis severity.

## 8. Discussion and Conclusions

In this article, a narrative review of the available information on probiotics and related substances has been carried out. In general, most studies indicate that probiotics could be an interesting therapeutic adjunct, as the data reveal beneficial effects on chemotherapy-induced mucositis.

Most of the studies in this review have been performed using animal models in rodents, and the number of antitumor drugs tested is limited; the use of 5-FU predominates [28,31,32,33,34,35,36,37,38,40,42,43,45,47,49,53,54,55,58,60,61,65,67,69,73,74,76,77,78,79], followed by other drugs such as MTX [44,50,51,57,59,68,72,80] or irinotecan [59,68,72,80]. Studies on chemotherapy based on platinum compounds [56,66] are very scarce, while other drugs such as doxorubicin [52] or capecitabine [30] have hardly been addressed. Furthermore, very few studies have analyzed the effects of multidrug combinations, which are common in cancer treatment [39,41,48,70,71]. In this aspect, it would be necessary to diversify the studies by increasing the number of drugs evaluated and making the protocols as similar as possible to those used in the clinic.

Within the administration protocols, most studies have only used a specific strain of probiotics, mainly *Lactobacillus* and *Bifidobacterium* genera (see Table 2) or mixtures of probiotics with the aim of combining the properties of several strains, including those (see Table 3). However, these probiotic mixtures seem to have similar effects to the use of isolated probiotics. Both alternatives seem to offer protection against gastrointestinal mucositis in several aspects, mainly by decreasing proinflammatory cytokines (such as TNF, IL-6, IL-1β and some others) [30,32,34,36,38,39,42,43,45,53,54,60,65,67,69]. Some of these studies reflect that these changes are related to the modulation of NF-κB expression [28,32,34,39,47], a key factor in the development of mucositis. Interestingly, some probiotics have demonstrated the ability to modulate the members of the TLR family, namely TLR-4 and TLR-2 [28,32,34,65]. Bacteria found in the gut are regulated by this family of receptors [81], which play a key role in the regulation of the nuclear factor NF-κB [28,32,34]. In some cases, the action of probiotics is not limited to reducing proinflammatory cytokines, but they also elevate the levels of anti-inflammatory cytokines such as IL-10 [42,43,53,69]. IL-10 is a very important cytokine in stopping the inflammatory response and preventing excessive immune responses [82], and some probiotics such as *Lactobacillus acidophilus* [42], *Lactoplantibacillus plantarum* [43], *Streptococcus thermopilus* [53], or some of their combinations [69] seem to increase it.

Probiotics and probiotic mixtures also have the capacity to attenuate the histopathological changes caused by the use of the drugs studied [30,33,36,39,42,43,44,45,47,49,53,55,57,58,60,67,68,69,70], causing experimental animals to recover the normal structure and function of the gastrointestinal wall and alleviating the alterations in gastrointestinal permeability that usually occur during mucositis [32,54,55,57,65]. They also correct the loss of mucin-secreting goblet cells [31,33,36,68], or the expression of genes encoding mucin secretion [32], thus maintaining their protective functions. Probiotic therapy also improves the intestinal immune barrier, mainly through the intestinal immunoglobulin A response [33,55].

Because of all the above, they also alleviate other factors related to the occurrence of mucositis such as weight loss and diarrhea both in animal models [30,31,33,36,38,39,41,43,45,49,52,60,66,67,68,69,70] and in clinical trials with patients [40,56,71,72].

Some of these studies have demonstrated the ability to modulate the alterations in the microbiota that occur during chemotherapy, favoring the growth of beneficial bacterial strains. Specifically, the probiotics *Lacticaseibacillus casei* [33], *Lacticaseibacillus rahnmosus* [37], *Limosilactobacillus reuteri* [37], *Bifidobacterium bifidum* [45], *Bifidobacterium breve* [46], *Clostridium butirycum* [56], or mixtures of these compounds such as DM#1 [65] have been shown to modulate these changes. It should be noted that the relationship between cancer and gut microbiome is complex. They can affect the growth and spread of cancer, detoxify dietary components, and reduce inflammation. Microbiota can influence carcinogenesis by altering host cell proliferation and death, guide immune system function, and influence the metabolism of factors produced by the host, ingested food, and drugs [83]. Therefore, this modulation could be important not only in alleviating mucositis, but also in influencing chemotherapy efficacy and disease progression. In relation to synbiotics, they are the least studied compounds of this type (see Table 4) and only in animal models. The results obtained are along the same lines as those obtained with probiotics and their mixtures without apparent improvements. Interestingly, in some studies they seem to have a controversial effect on constipation-related symptoms such as bloating, abdominal discomfort, or pain [84]. Whether synbiotics are a better option than probiotics is still unclear [85].

In any case, it is important to highlight that the effects of probiotics vary according to the experimental model, the strains selected, or the protocol. In this context, *Lacticaseibacillus rhanmosus* was not effective against mucositis produced by 5-FU in a Dark Agouti rat model [35], while in other animal models it did show efficacy [37,38,39]. The studies using *Limosilactobacillus fermentum* or *Bifidobacterium lactis* in the same animal model also failed [35]. The study of different subtypes of *Streptococcus thermophilus* showed that not all of them work to alleviate mucositis [53]. Furthermore, a study demonstrated the importance of the administered dose of this probiotic, since only the highest doses used alleviated the negative effects of mucositis [29]. Regarding the yeast *Saccharomyces boulardii*, it was able to alleviate the symptoms of mucositis caused by 5-FU administered at a dose of 450 mg/kg [28,60], but not at 300 mg/kg [61].This result suggests that not all probiotic regimens will be efficient in all patients, and, therefore, the probiotic regimen to be used in the clinics must be carefully selected according to the patient conditions and needs.

In addition, in animal models that mimicked the presence of the tumor during chemotherapeutic treatment, *S. thermophilus* was not sufficient to alleviate the mucositis produced by MTX [51] in Dark Agouti rats. Other probiotics such as *Lacticaseibacillus casei* [29,30] or the mixture of *Lactiplantibacillus plantarum* and *Streptococcus thermophilus* [69] did show efficacy in cancer models and were even capable of blocking tumor growth [29,69].

Few studies have been focused on the effects of introducing live bacterial strains into the body of cancer patients [86]. It is believed that the administration of these probiotics could have side effects on the organism such as systemic infections, detrimental metabolic activities, excessive immune stimulation in susceptible individuals, or even gene transfer [87], compromising the health of patients. Only one study has used immunosuppressed mice to mimic the state of some cancer patients, without finding evidence of bacteremia [38], and two have analyzed the possibility of bacterial translocation in animals treated with antitumor drugs [36,38]. Importantly, one of them [38] detected the presence of the probiotic mixture administered in other locations. The safety of administering probiotics to immunocompromised patients and the possibility that these bacteria may cause infections that compromise patient safety should be further studied. Thus, the use of probiotics and derivatives should be taken with caution and their own potential adverse effects should be carefully analyzed. In this regard, it may be necessary to extend studies using immunosuppressed animals that more adequately mimic the delicate situation of cancer patients undergoing chemotherapy. Studies in normal animals could be masking important aspects in this respect.

With this in mind, there are some studies using postbiotics (Table 5) or paraprobiotics (Table 6) to alleviate the symptoms of mucositis produced by chemotherapy. However, they are very few and limited; studies on postbiotics have only used 5-FU [49,78,79], and only one study on paraprobiotics with irinotecan has been found [80]. Although they seem to have some positive effects, they do not seem to be as effective as the administration of the live probiotic. Furthermore, these studies have been carried out in animals without tumor or immunosuppression, and so it is still necessary to prove that the use of these derivatives is advantageous in these conditions, compared with preparations based on live bacteria.

In conclusion, many studies suggest that different probiotics may alleviate or prevent the development of mucositis through different mechanisms of action (Table 7).

Nevertheless, the efficacy and safety of probiotics and probiotic-derived products in animal models bearing tumors and in cancer patients, which may be immunocompromised, has scarcely been studied.

Thus, despite the promising results obtained thus far (mainly using probiotics and mixtures of them), more studies are needed to clearly define the best way to use probiotics and probiotic-like agents as a therapy for the treatment of the side effects of chemotherapy in the gastrointestinal tract, particularly mucositis and its consequences on health, quality of life, and even survival.

## Figures and Tables

**Figure 1 jpm-13-01487-f001:**
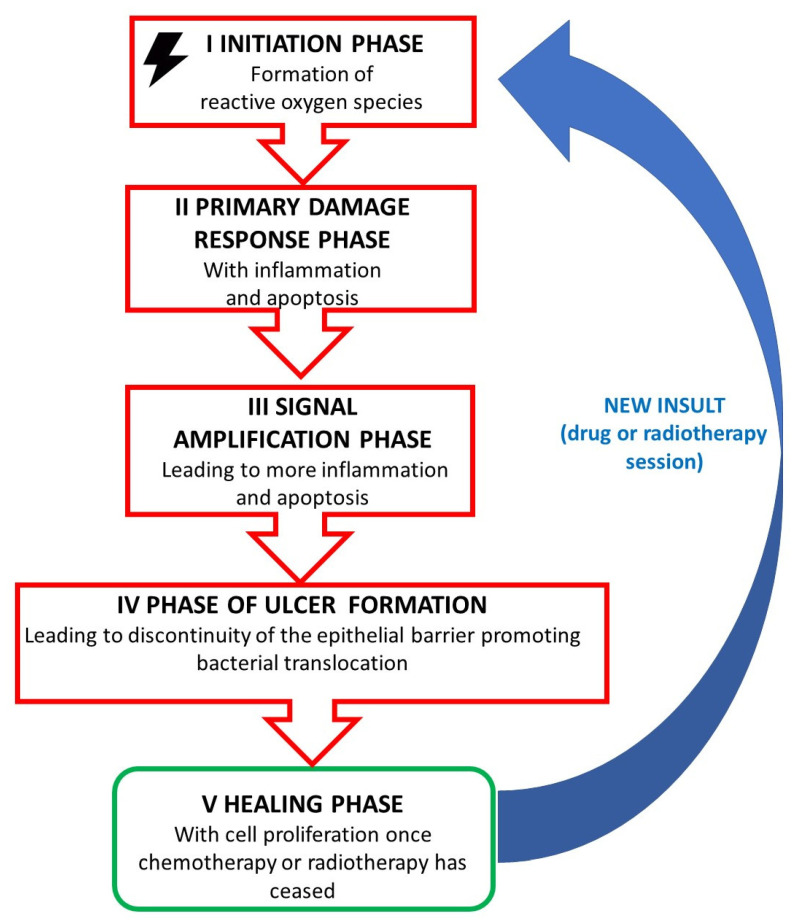
Five-step model defining pathogenesis of mucositis. Adapted from Touchefeu et al., 2014; Elad et al., 2020; McQuade et al., 2020 [13,14,15].

**Figure 2 jpm-13-01487-f002:**
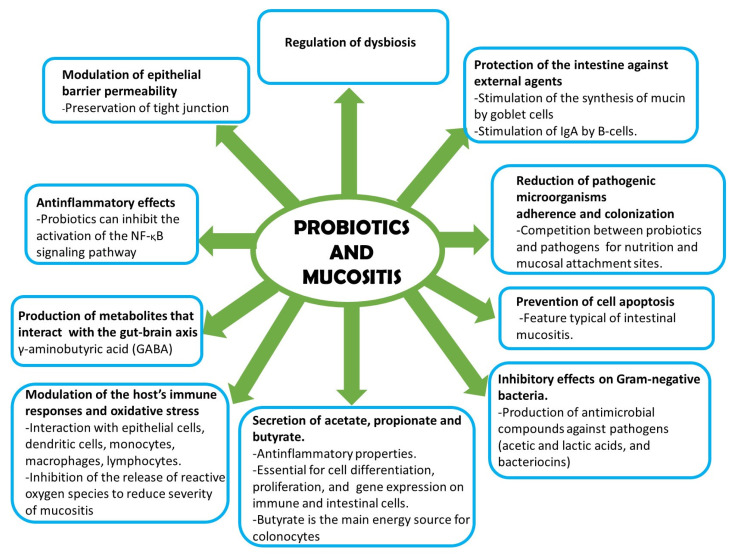
Main mechanisms of action of probiotics to improve intestinal mucositis. Adapted from Batista et al. [26].

**Table 1 jpm-13-01487-t001:** Definitions of the different relevant terms related to the use of probiotics and probiotic-related derivatives [19,20,21,22,23].

Probiotics	Live microorganisms which when administered in adequate amounts confer a health benefit on the host.
Synbiotics	A mixture of probiotics and microbial dietary supplements (prebiotics) that selectively stimulates the growth and/or activate the metabolism of one or a limited number of health-promoting bacteria, improving host welfare.
Postbiotics	Soluble factors originated from bacterial lysis or secreted by live bacteria
Paraprobiotics	Non-viable microbial cells or cell fractions that confer benefits when administered in adequate amounts
Probiotic Mixtures	Combinations of different probiotic bacteria, able to enhance the beneficial effects of isolated probiotics.

**Table 2 jpm-13-01487-t002:** Probiotics that have been studied for the treatment of chemotherapy-induced mucositis.

Probiotic	Experimental Model	Chemotherapy Drug	Main Findings	References
*Lacticaseibacillus casei*	Breast cancer model Balb/c mice	None	Delay or blocked tumor development. Modulation of the immune response.	[29]
	Breast cancer model Balb/c mice	Capecitabine	Decreases intestinal mucositis and mortality. Decreases diarrhea. Maintains the villi length/crypt depth ratio. Reduces cytokine IL-6.	[30]
	Balb/c mice	5-FU	Prevents weight loss and intestinal damage. Prevents degeneration of goblet cells.	[31]
	Swiss mice (female)	5-FU	Reduces inflammation in the colon and small intestine. Decreases TNF-α, IL-1β, IL-6, and malondialdehyde levels. Down-regulates NF-κB-P65 and TLR-4 gene expressions. Up-regulates MUC-2 and mucosal barrier proteins occludin and ZO-1 gene expressions. Normalizes intestinal microbiota. Unable to protect against 5-FU-induced weight loss.	[32]
*Lactobacillus* *delbrueckii*	Balb/c mice	5-FU	Reduces 5-FU-induced shortening of small intestine length. Recovers the loss in weight. Reduces the increased intestinal permeability. Restores the loss of architecture of the ileum mucosa. Reduces levels of neutrophil and eosinophil infiltration. Prevents loss of goblet cells. Decreases the levels of sIgA.	[33]
	Balb/c mice	5-FU	Modulates inflammatory pathways. Decreases the gene expression of TLR2, TLR-4, Myd88, and NF-κB1. Decreases mRNA expression of proinflammatory cytokines IL-6 and IL-1β. Up-regulates anti-inflammatory IL-10.	[34]
*Lacticaseibacillus rhanmosus*	Dark agouti rats	5-FU	The probiotic was not effective against mucositis.	[35]
	Balb/c mice	5-FU	Attenuates diarrhea developed by 5-FU. Repairs damage in jejunal villi. Prevents upregulation of TNF-α, IL-1β, and IL-6. Restores villus shortening, modification of crypt depth, and reduction in goblet cell number.	[36]
	Balb/c mice	5-FU	Modulation of microbiota composition. Increases the abundances of *Enterobacteriales* and *Turicibacterales*.	[37]
	Nod/scid mice	5-FU	Improves diarrhea scores. Inhibits TNF-α, IL-1β, and IL-6 levels. No evidence of bacteremia.	[38]
	Balb/c mice	FOLFOX	Reduces the severity of diarrhea and intestinal mucositis. Alleviates villus shortening, the lengthening of the intestinal crypts and reduces the villus height-to-crypt depth ratio. Inhibits NF-κB activity. Abrogates upregulation of TNF-α and IL-6. Reduces apoptosis of the intestinal crypt cells. Prevents changes in microbiota.	[39]
	Patients	5-FU	Less grade of diarrhea. Less abdominal discomfort.	[40]
	Patients	Vincristine, daunorubicin, L-asparaginase	Decreases nausea, vomiting, and abdominal distension.	[41]
*Lactobacillus acidophilus*	Swiss mice	5-FU	Restores the villus height–crypt depth ratio. Reduces glutathione concentration. Decreases MPO activity and nitrite concentrations in jejunum and ileum. Decreases cytokine concentrations. (TNF-α, IL-1β, and CXCL-1) Increases IL-10 concentrations. Restores delayed gastric emptying and gastrointestinal transit.	[42]
*Lactiplantibacillus plantarum*	Balb/c mice	5-FU	Attenuates weight loss, marked diarrhea, and shortening of villus height. Reduces proinflammatory cytokine concentrations (IL-17, TNF-α, INF-γ, and IL-6) in serum. Reduces IL-17, TNF-α, IFN-γ, IL-6, IL-4, and IL-2 in intestinal contents. Elevates the production of IL-10.	[43]
*Lactobacillus johnsonii*	Sprague–Dawley rats	MTX	Maintains intestinal integrity. Reduces histological damage. Not effective in restoring brush border enzymatic activity.	[44]
*Limosilactobacillus reuteri*	Balb/c mice	5-FU	Modulation of gut microbiota. Increases the abundances of *Enterobacteriales* and *Turicibacterales*.	[37]
*Limosilactobacillus fermentum*	Dark agouti rats	5-FU	Does not prevent the small intestine damage.	[35]
*Bifidobacterium bifidum*	ICR (CD1) mice	5-FU	Attenuates body weight loss and reduces the severity of diarrhea. Restores reduction in small intestine length, shortening of the villi, and destruction of the crypts. Decreases MPO activity and TNF-α and IL-1β expression. Partially restores normal composition of microbiota. Failed to prevent the initial induction of apoptosis.	[45]
*Bifidobacterium breve*	Pediatric patients with leukemia	Not specified	Increases *Clostridium leptum* subgroup. Enhances the habitation of anaerobes. Maintains favorable intestinal microflora.	[46]
*Bifidobacterium infantis*	Sprage-Dawley rats	5-FU	Restores villus height in jejunum. Increases expression of PCNA. Reduces expression of NF-κB and pro-inflammatory factors. Decreases MPO concentration.	[47]
	Sprage-Dawley rats	5-FU/oxaliplatin	Modulates the activity of immune cells. Suppresses Th1 and Th17 responses. Reduces upregulation of levels of Th1 cells and its cytokines (IL-2, IL-12, and IFN-γ). Down-regulates the level of T-bet. Decreases RORγt mRNA. Promotes Foxp3^+^ Treg responses.	[48]
*Bifidobacterium lactis*	Dark agouti rats	5-FU	No protection against mucositis	[35]
*Streptococcus thermophilus*	Female Dark Agouti rats	5-FU	Partially reduces disease severity score. Decreases mitotic activity and reduces crypt fission. Potential to combat neoplasia.	[49]
	Female Dark Agouti rats	MTX	High dose partially prevents the loss of bodyweight, the decrease in food intake. High dose partially attenuates mucositis measured by SBT. Low doses offer no protection.	[50]
	Female Dark Agouti rats (tumor bearing)	MTX	No protection. Insufficient to diminish severity of mucositis in the presence of a tumor.	[51]
	Female Dark Agouti rats	Doxorubicin	Partially prevented the loss of body weight. Mechanisms related with folate production. Increases polymorphonucleocytes and lymphocytes in saline-treated animals receiving TH-4.	[52]
	Female Balb/c mice	5-FU	Prevents intestinal mucositis, decreases diarrhea scores. Improves histological alterations and reduces jejunal inflammation. Decreases pro-inflammatory IL-6, increases anti-inflammatory IL-10. Lack of a beneficial effect of *S. thermophilus* CRL 415 vs. CRL 808.	[53]
	Balb/c mice	5-FU	Alleviates diarrhea. Decreases the infiltration of macrophages. Protects the structural integrity of small intestine and colon. Increases fecal acetic acid concentration. Reduces proinflammatory cytokines TNF-α, IL-1β, IL-6. Improves gut permeability.	[54]
*Propionibacterium freudenreichii*	Balb/c mice/HT-29 cells	5-FU	Reduces expression of proinflammatory cytokines IL-8 and TNF-α in LPS-stimulated HT-29 cells. Prevents weight loss. Reduces inflammation, and histopathological scores. Restores claudin-1 expression. Decreases sIgA levels. Mutation of the extractable surface protein slpB affects the probiotic effect.	[55]
	Balb/c mice	5-FU	Decreases intestinal inflammation. Preserves mucosal integrity. Reduces weight loss. Presence of a protective matrix does not improve its effects	[31]
*Clostridium butirycum*	Patients	Platinum based	Decreases chemotherapy-induced diarrhea. Decreases systemic inflammatory response. Helps to maintain the condition of the intestinal flora. Increases bacterial genera producing SCFAs. Reduces pathogenic genera. Increases beneficial flora, including *Clostridium* and *Lacticaseibacillus*.	[56]
*Saccharomyces cerevisiae*	Swiss mice	Irinotecan	Only post-treatment with viable bacteria offers protection. Prevents loss of goblet cells in animals with mucositis. Improves architecture of intestinal mucosa. Reduces mucosal inflammation. Reduces the alteration of intestinal permeability. Reduces damage caused by the formation of ROS.	[57]
	Swiss mice	5-FU	Reduces intestinal permeability. Reduces MPO activity. Reduces neutrophil chemoattractant cytokine CXCL1/KC. Decreases histopathological tissue damage and oxidative stress. Prevents the increase in NO levels. Selenium-enriched yeast reduces eosinophil peroxidase activity.	[58]
*Saccharomyces boulardii*	Sprague–Dawley rats	Irinotecan	Improves mucositis. Reduces leukocyte migration and inflammation. Modifies the signaling pathways implicated in the synthesis of proinflammatory cytokines.	[59]
	Swiss mice	5-FU	Enhances intestinal transit and gastric emptying. Attenuates diarrhea and weight loss. Reduces neutrophil infiltration, MPO activity, nitrite concentration, GSH concentration. Reduces proinflammatory cytokines TNF-α and IL-1β. Decreases apoptosis in intestinal crypt cells in the jejunum and ileum.	[60]
	Swiss mice/Caco cells	5-FU	Modifies the Toll-like/MyD88/NF-κB/MAPK pathway.	[28]
	Swiss mice	5-FU	Not able to prevent the effects of experimental mucositis.	[61]

Abbreviations: 5-FU, 5-fluorouracil; CXCL-1, chemokine (C-X-C motif) ligand 1; Foxp3, forkhead box P3; GSH, glutathione; IFN-γ, interferon gamma; IL, interleukin; KC, keratinocyte-derived chemokine; LPS, lipopolysaccharide; MAPK, mitogen-activated protein kinase; MPO, myeloperoxidase; mRNA, messenger ribonucleic acid; MTX, methotrexate; MUC-2, mucin-2; Myd88, myeloid differentiation primary response 88; NF-κB, nuclear factor kappa B; NO, nitric oxide; PCNA, proliferating cell nuclear antigen; RORγt, RAR-related orphan receptor gamma; ROS, reactive oxygen species; SBT, sucrose breath test; SCFA, short chain fatty acid; sIgA, secretory immunoglobulin A; Th, T helper; TH-4, *Streptococcus thermophilus*; TLR, toll-like receptor; TNF-α, tumor necrosis factor alpha; Treg, regulatory T cells; ZO-1, zonula ocludens-1.

**Table 3 jpm-13-01487-t003:** Probiotic mixtures that had been studied for the treatment of chemotherapy-induced mucositis.

Probiotic Mixture	Experimental Model	Chemotherapy Drug	Main Findings	References
*Lactobacillus bulgaricus/Streptococcus thermophilus*	Sprague–Dawley rats	MTX	Improves brush-border sucrase and lactase activity, and tissue architecture. Improves small intestinal barrier function.	[44]
Probiotic mixture, DM#1	Sprague–Dawley rats	5-FU	Reduces levels of proinflammatory cytokines TNF-α and IL-4 and IL-6. Reduces neutrophil infiltration in the intestine. Decreases gut permeability and plasma endotoxin concentrations. Restores *Lacticaseibacillus* spp. and *Clostridium* clusters III and XIVa. Attenuates the increase in TLR-2 mRNA expression.	[65]
Probiotic mixture, DM#1	Sprague–Dawley rats	Cisplatin	Ameliorates cisplatin-induced mucositis and pica. Normalizes the dysbiosis-driven 5-HT overproduction.	[66]
PM1-*Lactobacillus acidophilus/Bifidobacterium lactis*	Swiss mice	5-FU	Increases the villus height/crypt depth ratio of the intestinal mucosa. Decreases histopathological scores. Reduces GSH consumption in the duodenum and jejunum. Reverses the increased lipid peroxidation. Reduces neutrophil infiltration and cytokines TNF-α and IL-6. Does not prevent loss of body weight.	[67]
PM2-*Lactobacillus acidophilus, Lacticaseibacillus paracasei, Lacticaseibacillus rhamnosus, Bifidobacterium lactis*	Swiss mice	5-FU	Prevent loss of body weight. Increases the villus height/crypt depth ratio of the intestinal mucosa. Decreases histopathological scores. Reduces GSH consumption in the duodenum and jejunum. Reverses the increased lipid peroxidation. Reduces neutrophil infiltration and cytokines TNF-α and IL-6. Reverses the delay in gastric emptying.	[67]
Labi-*Lactobacillus acidophilus/Bifidobacterium bifidum*	Balb/c mice	5-FU	Attenuates diarrhea development. Suppresses the upregulation of TNF-α, IL-6 and IFN-γ. Restores villus height. No bacteremia.	[36]
Labi-*Lactobacillus acidophilus/Bifidobacterium bifidum*	Nod Scid mice	5-FU	Attenuates diarrhea development. Suppresses the upregulation of TNF-α, IL-6 and IFN-γ. Does not restore villus height. Labi strains detected in the spleen, liver samples, and mesenteric lymph nodes.	[38]
Probiotic mixture VSL#3	Female Dark Agouti rats	Irinotecan	Reduces weight loss. Prevents moderate and severe diarrhea. Increases epithelial proliferation and healing of the mucosal layer in jejunum and colon. Reduces intestinal apoptosis. Prevents the increase in goblet cell.	[68]
*Lact. plantarum* CRL 2130, *Strep. thermophilus* CRL 808 and *Strep. thermophilus* CRL 807	Balb/c mice (model of breast cancer)	5-FU	Decreases tumor growth. Reduces damage at intestinal structure level. Reduces diarrhea degree. Decreases serum concentrations of cytokines IL-6 and TNF-α. Increases levels of the regulatory cytokine IL-10.	[69]
*Lacticaseibacillus casei* W56; *Lactobacillus acidophilus* W37; *Levilactobacillus brevis* W63; *Lactococcus lactis* W58; *Bifidobacterium lactis* W52; *Lactococcus lactis* W19; *Ligilactobacillus salivarius* W24; and *Bifidobacterium bifidum* W23	Wistar rats	FOLFOX	The occurrence and severity of diarrhea was controlled. Alleviates the intestinal damage induced by chemotherapy. Increases intestinal cell proliferation. Reduces apoptosis of cells in the ileum, but not in colon samples.	[70]
*Bifidobacterium tetragenous*	Patients	CHOP for patients with lymphoma, fluoropyrimidine-based chemotherapy for patients with colorectal cancer, TP regimen in patients with lung cancer	Constipation was controlled by the probiotic mixture.	[71]
Probiotic formula Colon Dophilus™	Patients	Irinotecan	Reduces the incidence of diarrhea (grade 3 or 4). Reduces the overall incidence of diarrhea. Reduces the incidence of enterocolitis.	[72]

Abbreviations: 5-FU, 5-fluorouracil; 5-HT, 5-hydroxytryptamine or serotonin; CHOP, cyclophosphamide, doxorubicin hydrochloride (hydroxydaunorubicin), vincristine sulfate (Oncovin), and prednisone; GSH, glutathione; IFN-γ, interferon gamma; IL, interleukin; mRNA, messenger ribonucleic acid; MTX, methotrexate; PM, probiotic mixture; TNF-α, tumor necrosis factor alpha; TP regimen, docetaxel plus cisplatin.

**Table 4 jpm-13-01487-t004:** Synbiotics that have been studied for the treatment of chemotherapy-induced mucositis.

Synbiotic	Experimental Model	Chemotherapy Drug	Main Findings	References
*Limosilactobacillus fermentum* BR11 and the prebiotic, fructo-oligosaccharide	Female Dark Agouti rats	5-FU	High dose of the synbiotic increases the severity of 5-FU-induced mucositis. Synbiotic does not provide additional protection than the probiotic alone.	[73]
Mulberry leaf extract and *Lactobacillus acidophilus* A4	Wistar rats	5-FU	Ameliorates 5-FU-induced weight loss. Ameliorates reduction of villi height. Reduces histological severity. Reduces IL-1β expression, and MPO activity. Reduces neutrophil infiltration in the small and large intestine. Stimulates expression of mucin genes and mucin production.	[74]
Synbiotic Gut Working Tablet	Sprague–Dawley rats	MTX	Reverses villi shortening, crypt loss, and goblet cell depletion in the mucosa. Reduces activities of MPO and lipid peroxidase. Increases superoxide dismutase activity. Suppresses mRNA expression of TNF-α and IL-12 in macrophages. Decreases TNF-α mRNA expression.	[75]
Simbioflora^®^	Balb/c mice	5-FU	Thicker mucus layer Maintains villus and crypt height. Reduces inflammatory infiltration in small intestine. Reduces EPO activity. Increases SCFAs acetate, butyrate, and propionate in the feces. The synbiotic does not reverse the increased MPO activity in the small intestine.	[76]
*Lacticaseibacillus casei* BL23, *Lactiplantibacillus plantarum* B7, *Lacticaseibacillus rhamnosus*, whey protein and fructo-oligosaccharide	Female Balb/c mice	5-FU	Anti-inflammatory activity. Reduces animal weight loss. Reduces intestinal permeability. Controls levels of pro-inflammatory cytokines. IL-1β, IL-6, IL-17, and TNF-α.	[77]

Abbreviations: 5-FU, 5-fluorouracil; EPO, eosinophil peroxidase; IL, interleukin; MPO, myeloperoxidase; mRNA, messenger ribonucleic acid; MTX, methotrexate; SCFAs, short-chain fatty acids; TNF-α, tumor necrosis factor alpha.

**Table 5 jpm-13-01487-t005:** Postbiotics that have been studied for the treatment of chemotherapy-induced mucositis.

Postbiotic	Experimental Model	Chemotherapy Drug	Main Findings	References
Supernatant from *Streptococcus thermophilus* TH-4	Female Dark Agouti rats	5-FU	Increases crypt depth. Improves crypt fission.	[49]
Supernatants from *Escherichia coli Nissle* 1917	Female Dark Agouti rats	5-FU	Reduces histological severity scores. Improves intestinal morphology. Reduces levels of MPO.	[78]
Supernatants from *Limosilactobacillus fermentum* BR11	Female Dark Agouti rats	5-FU	Reduces histological severity scores. Improves intestinal morphology. Increases MPO levels in the jejunum	[78]
Supernatants from *Faecalibacterium prausnitzii*	Female Dark Agouti rats Cell cultures	5-FU	Enhances barrier integrity and prevents the barrier disruption in Caco-2 and T-84 cells. Prevents body weight loss and normalizes water intake. Normalizes crypt depth in the jejunum. Fails to alleviate the increased MPO levels.	[79]
Supernatants from *Escherichia coli Nissle* 1917	Female Dark Agouti rats Cell cultures	5-FU	Prevents body weight loss and normalizes water intake. Normalizes crypt depth in the jejunum. Fails to alleviate the increased MPO levels.	[79]

Abbreviations: 5-FU, 5-fluorouracil; MPO, myeloperoxidase.

**Table 6 jpm-13-01487-t006:** Paraprobiotics that have been studied for the treatment of chemotherapy-induced mucositis.

Paraprobiotic	Experimental Model	Chemotherapy Drug	Main Findings	References
Paraprobiotic EC-12S	C57BL/6 male mice	Irinotecan	Attenuates the neutrophil infiltration. Increased the villus/crypt ratio. Reduces the mRNA expression of Cldn-2, Ocln. Inhibits irinotecan-induced bacterial translocation to the blood.	[80]
Paraprobiotic-based preparation Med Lan–S^®^	C57BL/6 male mice	Irinotecan	Attenuates the neutrophil infiltration. Increased the villus/crypt ratio. Reduces the mRNA expression of Cldn-2, Ocln, and TLR4. Inhibits the irinotecan-induced bacterial translocation to the blood.	[80]

Abbreviations: Cldn-2, claudin-2; mRNA, messenger ribonucleic acid; Ocln, ocludin; TLR, toll-like receptor.

**Table 7 jpm-13-01487-t007:** Mechanisms of action described for specific probiotics (at the species level).

Regulation of dysbiosis	*Lacticaseibacillus casei* *Lacticaseibacillus rhamnosus* *Bifidobacterium bifidum* *Bifidobacterium breve* *Clostridium butyricum*
Modulation of epithelial barrier permeability	*Lacticaseibacillus casei* *Lactobacillus delbrueckii* *Propionibacterium freudenreichii* *Saccharomyces cerevisiae*
Anti-inflammatory effects	*Lacticaseibacillus casei* *Lacticaseibacillus rhamnosus* *Lactobacillus delbrueckii* *Bifidobacterium bifidus* *Bifidobacterium infantis* *Streptococcus thermophilus* *Propionibacterium freudenreichii* *Clostridium butyricum* *Saccharomyces boulardii*
Modulation of host immune response	*Lactobacillus delbrueckii* *Lactobacillus acidophilus* *Bifidobacterium bifidum* *Bifidobacterium infantis* *Streptococcus thermophilus* *Propionibacterium freudenreichii* *Saccharomyces cerevisiae* *Saccharomyces boulardii*
Reduction of oxidative stress	*Saccharomyces cerevisiae* *Saccharomyces boulardii*
Prevention of apoptosis	*Saccharomyces boulardii* *Lacticaseibacillus rhamnosus*

## Data Availability

Not applicable.

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
