# Peer review of "Probiotics and Probiotic-like Agents against Chemotherapy-Induced Intestinal Mucositis: A Narrative Review"

_jpm, 2023, doi:10.3390/jpm13101487_

Round 1
Reviewer 1 Report
Dear Authors
1. Please define the inclusion/exclusion criteria of chosen articles
2. The method of study is not explained.
The English language is good enough.
Author Response
We would like to thank the reviewer for his positive consideration towards our manuscript and for his valuable suggestions that have helped us to improve our manuscript. We will answer both his questions in the following paragraphs, in which we describe the search method of our study, including the inclusion/exclusion criteria used for choosing the articles.
This is a narrative review that aims to gather all the available information on probiotics and related compounds, which in recent years have aroused great interest in the prevention of the side effects of chemotherapy; for that reason, inclusion and exclusion criteria have not been applied as strictly as in systematic reviews.
For the search of information, we have used the specialized health sciences database Pubmed, including both preclinical studies, using laboratory animals or cell cultures, and human trials, with no date limit and considering both positive and negative results regarding the effectiveness of the probiotic.
As exclusion criteria, we have eliminated all articles that did not specifically deal with gastrointestinal mucositis (for example, those about oral mucositis, which are abundant) and we have focused only on chemotherapy treatments, excluding other methods for the treatments of cancer such as radiotherapy or immunotherapy.
We have included all this information in the text, in a new section for methodology (lines 167-178) and we hope our manuscript is now satisfactory.
Reviewer 2 Report
This research article is excessively long and causes me some confusion when I read it. I have not visualized the methodology that has been followed in the selection of the articles to be reviewed. Nor have I read the discussion of the different articles. When I read the research question and review the conclusions in the first paragraph, “As shown above, the studies that have been carried out with probiotics are very diverse, the doses administered vary among studies, the animal models used are also different, as well as the chemotherapy drugs used, and the mucositis induction protocols (although 5-FU alone or combined with other drugs has probably been the most frequently used). Despite this heterogeneity the results are promising” this are more limitations of this review work than real conclusions.
Possibly I think that this research work could be a narrative review (it should appear in the title), where the authors intend to give a broad view on the subject, but at the same time propose new lines of research. For example, in view of the results, which probiotic strain or strains (postbiotics or paraprobiotics) would be susceptible to be used in a clinical trial to evaluate their efficacy? I understand, without a selection methodology and without discussing the results, it is quite complicated.
For all the above reasons, I believe that the authors should reconsider their research work and at least show how the results obtained agree or disagree with the objectives proposed in this article.
On the other hand, there are some typos or minor points that need to be corrected.
Minor points:
Authors should check that, when there is more than one citation, these should be separated by commas, but if they are correlative, the first and the last are mentioned separated by a hyphen (line 67).
Authors should check that, in all citations (Vancouver style) where works with more than one author are cited, the year is not written. Example: Batista et al. [26] not Batista et al., (2020) [26].
There is a current reclassification of the family Lactobacillaceae into some Lactobacillus genera (Zheng et al., 2020), but not all have been renamed. For example: Lactobacillus delbrueckii subsp. delbrueckii has not changed to Lacticaseibacillus delbrueckii. Neither has Lactobacillus johnsonii changed to Lacticaseibacillus johnsonii. However, Lactobacillus reuteri has been renamed as Limosilactobacillus reuteri, but not as Lacticaseibacillus reuteri. Authors should review the taxonomy of all probiotic species.

Author Response
This research article is excessively long and causes me some confusion when I read it. I have not visualized the methodology that has been followed in the selection of the articles to be reviewed.
Thank you very much for your review and comments, we have taken them into account to try to improve the article following your recommendations. We hope that the changes we have made are appropriate and improve the quality of the text.
In line with your comments, we have included in the text a new section for methodology (lines 167-178), and we have tried to summarize the text and reduce its length to make it more understandable.
Nor have I read the discussion of the different articles. When I read the research question and review the conclusions in the first paragraph, “As shown above, the studies that have been carried out with probiotics are very diverse, the doses administered vary among studies, the animal models used are also different, as well as the chemotherapy drugs used, and the mucositis induction protocols (although 5-FU alone or combined with other drugs has probably been the most frequently used). Despite this heterogeneity the results are promising”this are more limitations of this review work than real conclusions. Possibly I think that this research work could be a narrative review (it should appear in the title), where the authors intend to give a broad view on the subject, but at the same time propose new lines of research. For example, in view of the results, which probiotic strain or strains (postbiotics or paraprobiotics) would be susceptible to be used in a clinical trial to evaluate their efficacy? I understand, without a selection methodology and without discussing the results, it is quite complicated. For all the above reasons, I believe that the authors should reconsider their research work and at least show how the results obtained agree or disagree with the objectives proposed in this article.
According to your suggestions we have indicated in the title that the manuscript is a narrative review. In addition, as requested by the reviewer, we have now included a new Discussion section in which the main results are highlighted and interpreted.
On the other hand, there are some typos or minor points that need to be corrected.
Minor points:
Authors should check that, when there is more than one citation, these should be separated by commas, but if they are correlative, the first and the last are mentioned separated by a hyphen (line 67). Authors should check that, in all citations (Vancouver style) where works with more than one author are cited, the year is not written. Example: Batista et al. [26] not Batista et al., (2020) [26].
Thank you very much for your comment, we have corrected all these minor points in the document.
There is a current reclassification of the family Lactobacillaceae into some Lactobacillus genera (Zheng et al., 2020), but not all have been renamed. For example: Lactobacillus delbrueckii subsp. delbrueckii has not changed to Lacticaseibacillus delbrueckii. Neither has Lactobacillus johnsonii changed to Lacticaseibacillus johnsonii. However, Lactobacillus reuteri has been renamed as Limosilactobacillus reuteri, but not as Lacticaseibacillus reuteri. Authors should review the taxonomy of all probiotic species.
Thank you very much for bringing this relevant mistake to our attention. We have checked the manuscript and adjusted taxonomy of all probiotic species according to Zheng et al, 2020, as required.
Reviewer 3 Report
The review entiteled "Probiotics and probiotic-like agents against chemotherapy-induced intestinal mucositis: a way to prevent further risks?" by Lopez-Gomez and colleagues is a very extensive overwiev of the vast field of probiotic related research. The present work presents a good overview of the recent literature.
Minor comment: Some information seems to be redundant (text, graphs, table). If the authors could tighten up the manuscript the quality of the review will improve.
Author Response
We would like to thank the reviewer for his positive consideration towards our manuscript and for his suggestions. We have improved the draft along the lines suggested and hope the reviewer finds the new version of our manuscript satisfactory.
We have tried to reduce the length of the text and eliminate repetitions by summarizing the information that already appears in Figures 1 and 2 and table 1, and directing the readers to the figures or tables that are more explanatory. However, at the request of another reviewer we have added two new sections, Methods and Discussion, which may have lengthened the article again. Despite this, the manuscript (from Introduction to Conclusions) has been shortened by almost 800 words. We hope it is now acceptable.
On the other hand, in relation to the other tables appearing in the manuscript (tables 2 to 6), it is true that they collect information that already appears in the text, but we believe they may serve as a guide and a summary to facilitate the text follow-up. In addition, these tables may allow the reader to find the information they need quickly without having to review the entire text. In a deep revision article with such varied contents, we believe these tables will be helpful for researchers who are looking for specific information about one of these products. We hope this is acceptable.
Round 2
Reviewer 1 Report
Dear Authors
The revised version is acceptable.
Reviewer 2 Report
I have just had a reading of the authors' replies and I am of the opinion that they are acceptable for publication of the article.